# Implicit Surface Reconstruction from Sparse and Noisy Poses with Large Motions

## Abstract

Recent advances in implicit surface reconstruction have significantly improved 3D reconstruction techniques. However, challenges persist, particularly when dealing with sparse and noisy poses. Traditional methods attempted to address these challenges through photometric and geometric consistency, but they often struggled as camera baselines increased. This difficulty arises due to incorrect guidance caused by occlusions during the learning of neural implicit representation. To overcome this issue, we propose an approach that incorporates uncertainty-aware guidance for multi-view consistency, allowing for better adaptation to scenarios with sparse and noisy inputs. Additionally, to facilitate the learning of surface geometry in a challenging setup, we propose a geometric smoothing termed progressive SDF loss. Through empirical studies on occlusion handling and geometric smoothing, our method achieved state-of-the-art performance, significantly enhancing both the refinement of noisy camera poses and surface reconstruction quality. This advancement strengthens the robustness and flexibility of implicit surface reconstruction in challenging conditions, paving the way for more effective applications in computer vision and 3D scene understanding.

## 1 Introduction

In 3D surface reconstruction, implicit surface representations based on neural radiance fields (Mildenhall et al., 2021) and signed distance functions (SDF) have delivered impressive results (Yariv et al., 2020; Wang et al., 2021a; Yariv et al., 2021). By leveraging the inherent properties of SDF to implicitly define surfaces, these methods exhibit remarkable versatility in handling challenges such as detailed surface reconstruction and complex surface topologies. As a result, they have excelled in producing accurate geometries from multi-view images, gaining widespread success in 3D surface reconstruction applications (Takikawa et al., 2021; Li et al., 2023; Zhu et al., 2024; Raj et al., 2025). However, relying on a large number of images and highly accurate camera poses remains a significant limitation. SCNeuS (Huang et al., 2024c) has further advanced neural surface reconstruction techniques by focusing on the challenging cases of sparse and noisy camera poses. By proposing a differentiable on-surface intersection for fast sampling and incorporating it into view-consistency loss, SCNeuS significantly enhances the potential of implicit surface methods in complex scenario. However, this approach lacks occlusion handling for view-consistency, resulting in decreased effectiveness with large camera baselines.

We propose an approach for refining camera poses in sparse-view settings that incorporates occlusion handling. Although previous methods use generalization (Long et al., 2022; Ren et al., 2023; Na et al., 2024), information gain regularization (Kim et al., 2022), or correspondence matching (Truong et al., 2023), we empirically found that a patch-wise photometric consistency loss (Darmon et al., 2022) effectively refines camera poses. To further improve performance in large baseline settings, we introduce an occlusion-handling approach that calculates uncertainty based on the discrepancy of back-projected 3D coordinates from warped 2D pairs and a simple geometric consistency loss to enhance geometric understanding. Additionally, we propose a progressive SDF loss, which helps build the surface prior and allows the model to construct surfaces even with sparse and noisy poses.

In this paper, we demonstrate that the proposed methods achieve remarkable performance in 3D surface reconstruction under challenging conditions. This is accomplished by effectively handling

Figure 1: For a small camera baseline, (a) SCNeuS (Huang et al., 2024c), (b) NeuS (Wang et al., 2021a) + SPARF (Truong et al., 2023), and (c) our method successfully reconstruct surfaces for (d) the target view. However, as the baseline increases, the performance of (a) and (b) dramatically decreases due to limited information gathering in sparse views. In contrast, our method maintains strong performance by leveraging uncertainty-aware consistency and geometric smoothing.

occlusion to improve view-consistency losses and introducing a progressive SDF (pSDF) loss to learning geometry stably with a coarse-to-fine manner.

Our contributions can be summarized as follows:

- Address the challenges of implicit surface reconstruction with large camera baselines.
- Introduce a method to enhance the camera pose refinement and geometric understanding by simply incorporating geometry-based uncertainty.
- Propose a method for building surfaces in harsh conditions with a progressive SDF loss.
- Achieve remarkable performance in surface reconstruction, under challenging conditions.

## 2 RELATED WORKS

**Surface reconstruction** has been a long-standing research area in computer vision, aiming to create a continuous 3D surface from discrete 3D data such as point clouds (Bernardini et al., 1999; Lorensen & Cline, 1998; Kazhdan et al., 2006; Kazhdan & Hoppe, 2013). Classical approaches, including a ball-pivoting algorithm (Bernardini et al., 1999), Poisson surface reconstruction (Kazhdan et al., 2006; Kazhdan & Hoppe, 2013), and marching cubes (Lorensen & Cline, 1998) achieved significant success in reconstructing surfaces from point clouds, particularly impacting fields such as 3D modeling and environmental scanning. More recently, NeRF (Mildenhall et al., 2021) was introduced for novel view synthesis. Building on powerful characteristics of NeRF, NeuS (Wang et al., 2021a) and VolSDF (Yariv et al., 2021) enabled implicit surface reconstruction by bridging the gap between a density representation of NeRF and SDF. To further enhance performance, NeuralWarp (Darmon et al., 2022) and GeoNeuS (Fu et al., 2022) incorporated a traditional view-consistency technique, NCC loss (Schönberger et al., 2016; Xu & Tao, 2019). An alternative direction is leveraging dense priors, such as depth maps, as seen in MonoSDF (Yu et al., 2022) and Sparis (Wu et al., 2025).

**NeRF with sparse and noisy input views** has faced a challenge to achieve reliable results. Since NeRF requires numerous input images with accurate camera poses, there are two primary limitations, (i) the need for a large amount of data and (ii) the requirement for accurate camera poses. To solve (i), IBRNet (Wang et al., 2021b) and MVSNet (Chen et al., 2021) proposed a generalizable novel view synthesis by extracting features from given images. RegNeRF (Niemeyer et al., 2022) and InfoNeRF (Kim et al., 2022) applied regularization and alternative methods to collect geometric information from sparse views for novel view synthesis. For implicit surface reconstruction, SparseNeuS (Long et al., 2022), VolRecon (Ren et al., 2023) and UFORecon (Na et al., 2024) also used generalizability handling approaches, while S-volsdf (Wu et al., 2023) introduced soft

consistency, NeuSurf (Huang et al., 2024b) employed geometric fields for global alignment and feature consistency for local geometry, and Spurfies (Raj et al., 2025) disentangled geometry and appearance, incorporating additional geometric guidance from a pretrained network for joint SDF and appearance reconstruction in sparse views. For the second limitation, dealing with NeRF with noisy views, BARF (Lin et al., 2021) and NeRF-- (Wang et al., 2021c) jointly update the model parameters and camera poses using a photometric loss. SiNeRF (Xia et al., 2022) and GARF (Chng et al., 2022) demonstrated that activation functions help refine noisy poses, while SCNeRF (Jeong et al., 2021) introduced joint optimization of both extrinsic and intrinsic camera parameters. More recently, SPARF (Truong et al., 2023) addressed both limitations of NeRF by leveraging matching correspondences and augmented view geometric consistency. SCNeuS (Huang et al., 2024c) demonstrated surface reconstruction with sparse and noisy poses by incorporating patch-wise NCC loss and on-surface sampling. However, we found that SCNeuS struggles with large camera motion under noisy poses due to occlusions.

To overcome the limitations, we present an occlusion-handling for photometric and geometric consistency to refine camera pose, as well as a surface smoothing algorithm to construct reliable geometry with inaccurate camera poses.

## 3 METHOD

### 3.1 PRELIMINARY

**NeRF** (Mildenhall et al., 2021) was the first method to leverage multi-layer perceptrons (MLPs) to jointly encode both appearance and scene geometry. Given a ray $r(\mathbf{o}, \mathbf{d}) \in \mathcal{R}$, where $\mathbf{o} \in \mathbb{R}^3$ is the camera origin and $\mathbf{d} \in \mathbb{R}^3$ is the viewing direction, NeRF samples query points $\{\mathbf{P}_i = \mathbf{o} + t_i \mathbf{d} | i = 1, ..., n, t_i < t_{i+1}\}$. For the query points and the viewing direction, MLPs predict both a color $c_i$ and a density $\sigma$: $[\mathbf{c}; \sigma] = \mathrm{MLP}(\mathbf{P}_i, \mathbf{d})$. The predicted values are used to render the color $C_r$ of the ray $r$ via differentiable volumetric rendering:

$$C_r = \sum_{i=1}^{N} T_i \alpha_i \mathbf{c}_i \ \text{ where } \ \alpha_i = 1 - \exp(-\sigma_i \delta_i), \tag{1}$$

where $T_i$ is the transmittance defined as $T_i = \exp\left(-\Sigma_{j=1}^{i-1} \sigma_j \delta_j\right)$, and $\delta_i = t_{i+1} - t_i$ denotes the distance between adjacent sampled query points.

**NeuS** (Wang et al., 2021a) extends the NeRF framework by incorporating a signed distance function $s(\cdot)$ to represent the scene geometry, where the surface $\mathcal{S}$ is implicitly defined as the zero-level set of the SDF, i.e., $\mathcal{S} = \{\mathbf{P} \in \mathbb{R}^3 \mid s(\mathbf{P}) = 0\}$. Instead of using density $\sigma$, NeuS learns an SDF $s(\mathbf{P}_i)$ and computes the opaque $\rho(i)$ at point $\mathbf{P}_i$ as:

$$\rho(i) = \max\left(\frac{-\frac{d\Phi_s}{dt}(s(\mathbf{P}_i))}{\Phi_s(s(\mathbf{P}_i))}, 0\right). \tag{2}$$

where $\Phi_s$ is a sigmoid function. This can derive the opacity $\alpha_i$ as $1 - \exp(-\int_{t_i}^{t_{i+1}} \rho(t)dt)$, and can render the color by using Eq. 1 as the definition of transmittance $T_i$ as $\Pi_{j=1}^{i-1}(1 - \alpha_j)$. Furthermore, NeuS derives the surface normal $n(\mathbf{P})$, which can be represented as a gradient of the SDF value with respect to $\mathbf{P}$, as shown:

$$n(\mathbf{P}) = \frac{\partial s(\mathbf{P})}{\partial \mathbf{P}}. \tag{3}$$

**The patch-wise NCC loss** (Furukawa & Ponce, 2009; Darmon et al., 2022) is introduced to enforce photometric consistency across views by comparing patches of images in multi-view stereo. The loss computes the normalized cross-correlation (NCC) loss for patches $\mathbf{s}_j \subset I_t$ and $\mathbf{s}_k \subset I_s$ for the target frame $I_t$ and source frame $I_s$, as its equation is:

$$L_{ncc}(\mathbf{s}_j, \mathbf{s}_k) = \frac{Cov(I_t(\mathbf{s}_j), I_s(\mathbf{s}_k))}{Var(I_t(\mathbf{s}_j))Var(I_s(\mathbf{s}_k))}, \tag{4}$$

where $Cov$ and $Var$ denote the covariance and variance of the patches, respectively.

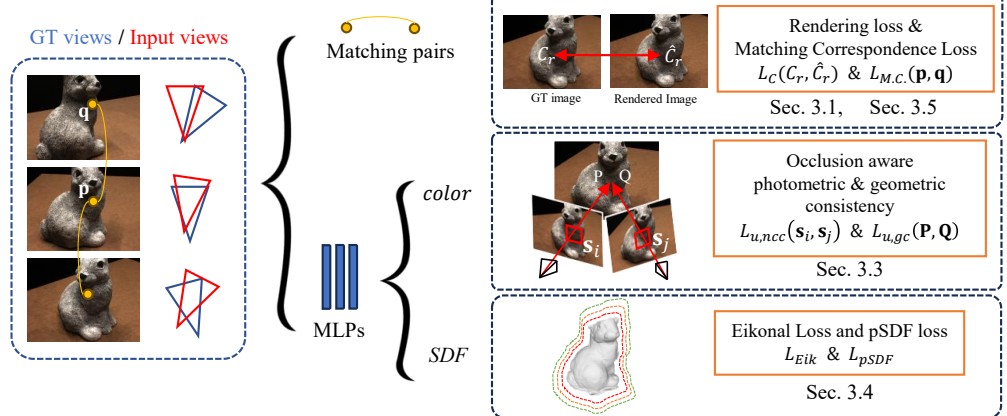

Figure 2: Our method optimizes a neural implicit representation with sparse, noisy camera poses by leveraging SDF properties. The model renders color and density from input views, using photometric loss (Sec. 3.1), matching correspondence loss (Sec. 3.5) and consistency losses (Sec. 3.3) for occlusion handling. The SDF values are used for the Eikonal loss and progressive SDF loss to construct implicit surfaces (Sec. 3.4).

## 3.2 OVERVIEW

Given an image set $\mathcal{I} = \{I_0, I_1, ..., I_m\}$, we extract correspondences using PDCNet (Truong et al., 2021) and sample rays and points with sparse and noisy cameras. The sampled points are processed by MLPs $f$ and $g$ to generate color $\mathbf{c}$ and SDF value $\mathbf{s}$ via differentiable rendering. The rendered values are used to compute uncertainty and losses, including a rendering loss, uncertainty-aware photometric and geometric consistency losses, and a surface loss that is composed of the Eikonal loss and the pSDF loss, which smooths complex geometry for easier optimization. Following BARF (Lin et al., 2021), a coarse-to-fine positional encoding is applied to aid stable training. The overall pipeline is shown in Fig. 2.

## 3.3 OCCLUSION HANDLING

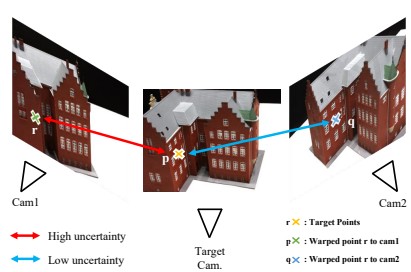

Figure 3: **Uncertainty**: As baselines of input views increase, uncertainty for warped patches $\mathbf{q}$ and $\mathbf{r}$ from $\mathbf{p}$ increases due to occlusions. The red line indicates high uncertainty caused by occlusions, while the blue line represents low uncertainty, corresponding to high accuracy and non-occluded regions.

We employ a simple, yet effective approach by enforcing geometric constraints that simultaneously refine camera poses and build implicit surfaces. As illustrated in Fig. 3, the uncertainty caused by occlusion is indicated by red (high) or blue (low). We compute the uncertainty $u \in \mathbb{R}^B$, where $B$ is a batch size, by representing it as a normalized distance between 3D points that are back-projected from 2D correspondences within a batch $\mathcal{B}$:

$$u_{\mathbf{p},\mathbf{q}} = \mathbf{N}(D_{\mathbf{p},\mathbf{q}}), \qquad (5)$$

where $\mathbf{N}(\cdot)$ denotes the min-max scaling function, and $D \in \mathcal{D}$ where $\mathcal{D}$ represents the set of squared Euclidean distances between the back-projected points. The points are derived from the 2D correspondences $\mathbf{p}$ and $\mathbf{q}$ from the target and reference frames, $I_t$ and $I_s$, respectively. The set of distances is defined as: $\mathcal{D} = \{|\pi^{-1}(\mathbf{p}, z_{\mathbf{p}}, P_t) - \pi^{-1}(\mathbf{q}, z_{\mathbf{q}}, P_s)|_2^2 \mid (\mathbf{p}, \mathbf{q}) \subset \mathcal{B}\}$. The inverse projection function $\pi^{-1}(\cdot, z, P)$ uses the per-pixel depth $z = \sum_{j=1}^n T_j \alpha_j t_j$ and the camera projection matrix $P$. To enhance performance in wide-baseline scenarios, we introduce an uncertainty-aware patch-wise NCC loss and an uncertainty-aware geometric consistency loss.

**Uncertainty aware patch-wise NCC loss**

SCNeuS (Huang et al., 2024c) used a straight-forward approach to adapt the NCC loss for the pose refinement, however, it is less effective in scenarios with large baselines due to occlusions. As shown in Fig. 3, occlusions induce high uncertainty, which makes learning challenging, especially with wide baselines. To address this issue, we incorporate the uncertainty value into the NCC loss simply as follows:

$$L_{u,ncc} = \frac{1}{B} \sum^{B} (\gamma_u \odot L_{ncc}), \quad (6)$$

where the $\gamma_u$ denotes a weighting factor for each patch as $\gamma_u = 1 - u$.

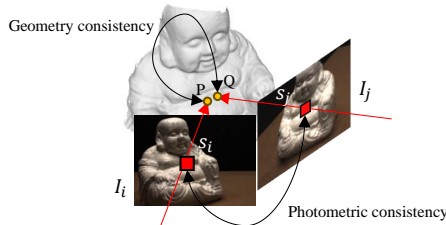

Figure 4: For photometric consistency, we applied a patch-wise NCC loss, using patch $\mathbf{s}_i$ in image $I_i$ and its warped patch $\mathbf{s}_j$ in image $I_j$ from $I_i$. A 3D distance between rendered points along each ray, $\mathbf{P}$ and $\mathbf{Q}$, are enforcing geometric constraints between the two 3D points.

**Uncertainty aware geometry consistency**

Truong et al. (2023); Kim et al. (2022) rely on augmented views for auxiliary geometry supervision, which increases training time by additional rendering processes. Instead, we establish strong and effective geometric consistency between warped points by leveraging the precomputed uncertainty:

$$L_{u,gc} = \frac{1}{B} \sum^{B} (\gamma_u \odot D). \quad (7)$$

Fig. 4 visualizes the occlusion-aware NCC loss and geometric consistency. This simplified approach improves model parameter updates efficiently.

### 3.4 IMPLICIT SURFACE SMOOTHING

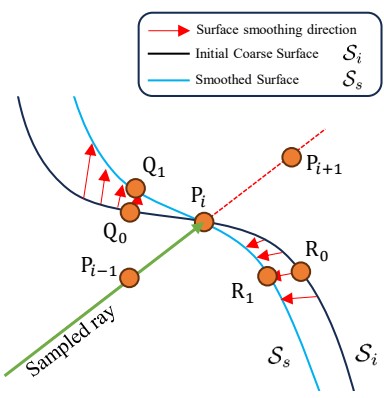

Figure 5: **Implicit surface smoothing**: Assume the closest point of $\mathbf{P}_{i-1}$ to the initial surface $\mathcal{S}_i$ is $\mathbf{Q}_0$. By enforcing the SDF value as the distance between point $\mathbf{P}_{i-1}$ and $\mathbf{P}_i$, the point $\mathbf{Q}_0$ is repelled to $\mathbf{Q}_1$. Conversely, the point $\mathbf{R}_0$ is attracted to $\mathbf{R}_1$.

Lack of information from the sparse input views complicates building an accurate surface due to errors from imperfectly optimized camera poses. To mitigate this difficulty, we introduce a progressive SDF loss. A similar method is introduced in iSDF (Ortiz et al., 2022), which uses batched sensor-based points to set SDF values as distances to the nearest sensor data, reducing noise and filling gaps. Instead of high-reliability sensor data, we leverage initial coarse geometry, optimized in early training, to sample on-surface points. The formulation is as follows:

$$L_{sdf} = (d' - s(\mathbf{P})) \cdot \mathbf{1}_{|d(\mathbf{P})-d|<b}, \quad (8)$$

where $d' = d - d(\mathbf{P})$, $\mathbf{1}$ is an indicator function, $d$ is the rendered depth value of a ray, $d(\mathbf{P})$ is the distance to $\mathbf{P} \in \mathbb{R}^3$ from the origin of the ray $\mathbf{o}$, and the $b$ is truncation region, which determines the affected regions by the SDF loss. An overview of the implicit surface smoothing is illustrated in Fig. 5. Let points $\mathbf{P}_{i-1}$, $\mathbf{P}_i$ and $\mathbf{P}_{i+1}$ be on the sampled ray from an optimizing camera. $\mathbf{P}_i$, $\mathbf{Q}_0$ and $\mathbf{R}_0$ are on the initial surface $\mathcal{S}_i$. In this case, the true SDF value for $\mathbf{P}_{i-1}$ is $\overline{\mathbf{P}_{i-1}\mathbf{Q}_0}$. However, by enforcing the SDF value as $\overline{\mathbf{P}_{i-1}\mathbf{P}_i}$ following Eq. 8, a conflict arises with the initial coarse surface $\mathcal{S}_i$. By enforcing this conflict, the point $\mathbf{Q}_0$ shifts to $\mathbf{Q}_1$, and, conversely, $\mathbf{R}_0$ moves to $\mathbf{R}_1$, creating a smoother surface $\mathcal{S}_s$. Intuitively, by forcing the SDF value of a sampled point as the distance between the point and the intersection of the ray and surface, true SDF value is repelled at concave surfaces and attracted at convex surfaces. However, applying SDF loss in the early training stage - before building the initial coarse geometry - can over-smooth surfaces and result into creating flat plane. To prevent this phenomenon, we apply our methods after initializing coarse geometry, specifically when the

high-frequency component of coarse-to-fine positional encoding becomes active. We then progressively shrink the truncation regions $b$ during optimization. Thus, the progressive SDF (pSDF) loss is defined as:

$$L_{pSDF} = h(k, b(k))L_{sdf}, \quad \text{if} \quad |d(\mathbf{P}) - d| < b, \tag{9}$$

where the $h(k, b(k))$ is a function to set a smoothing area, depending on the training iteration $k$ and the threshold value $b(k)$, which determines how many points are affected by Eq. 8. Details about the function $h(\cdot)$ are provided in the Appendix B.2.

### 3.5 OPTIMIZATION

To optimize our model, we utilize three types of losses: rendering loss, uncertainty-aware consistency losses, and surface reconstruction loss.

**Rendering loss** enforces the rendered color $C_r$ to match the ground truth color $\hat{C}_r$ for ray $r$ using a MSE loss. There is also a loss $L_{M.C.}$ for matching correspondences with a hyperparameter $\lambda_{M.C.}$.

$$L_C = \sum_{r \in \mathcal{R}} \|C_r - \hat{C}_r\|_2 + \lambda_{M.C.} L_{M.C.}, \text{ where } L_{M.C.}(\mathbf{p}, \mathbf{q}) = \sum_{(\mathbf{p}, \mathbf{q}) \in \mathcal{B}} \|\mathbf{p} - \pi(\pi^{-1}(\mathbf{q}, z_\mathbf{q}, P_s), P_t)\|_2 \tag{10}$$

where $\pi(\cdot, P)$ is a projection matrix with given camera matrix $P$.

**Uncertainty-aware consistency loss** consists of two components, photometric loss and geometric loss which are introduced in Eq. 6 and Eq. 7 with weighting hyperparameter $\lambda_{u,gc}$, defined as:

$$L_u = L_{u,ncc} + \lambda_{u,gc} L_{u,gc}. \tag{11}$$

**Surface reconstruction loss** comprises the Eikonal loss (Gropp et al., 2020) and our pSDF loss $L_{geo} = L_{Eik} + \lambda_{pSDF} L_{pSDF}$, where $L_{Eik} = \sum_{\mathbf{P} \in \mathcal{V}} (\|n(\mathbf{P})\| - 1)^2$ regularizes the surfaces, where $\mathcal{V}$ is a set of sampled points on the rays $r \in \mathcal{R}$.

Therefore, the total loss function $L = L_C + \lambda_u L_u + \lambda_{geo} L_{geo}$, where the $\lambda_u$ and $\lambda_{geo}$ represents the weighting parameters for each term.

## 4 EXPERIMENTS

### 4.1 IMPLEMENTATION DETAIL

**Architecture.** For our implementation, we used base architecture of NeuS (Wang et al., 2021a) and SPARF (Truong et al., 2023). To refine the camera poses, we hired a same strategy used in SPARF (Truong et al., 2023). For correspondence matching, PDCNet (Truong et al., 2021) is used.

**Evaluation.** To evaluate surface reconstruction and noisy camera pose refinement, we measured rotation error, translation error, and the Chamfer distance. To measure camera pose errors, we followed the evaluation strategy of SPARF. However, to calculate the Chamfer distance, a point cloud registration algorithm was utilized following Open3D (Zhou et al., 2018) due to freely optimized cameras. The detailed process is introduced in Appendix B.4. After post-processing, the Chamfer distance was measured following UniSurf (Oechsle et al., 2021) and IDRNet.

**Dataset and baselines.** We evaluated our method on the DTU dataset (Jensen et al., 2014) and the BlendedMVS dataset (Yao et al., 2020) under sparse and noisy view conditions, following the setup in SCNeuS. For the DTU dataset, we evaluated narrow-baseline (views 22, 23, and 24) and wide-baseline (views 22, 25, and 28) scenarios at a resolution of 1200×1600, utilizing 15 scans that provide 3D scene data, to validate camera pose accuracy and 3D reconstruction quality. For the BlendedMVS dataset, which lacks ground truth 3D models, we measure solely camera pose error using randomly selected 3 views at 768×576 resolution. Following BARF (Lin et al., 2021) and SPARF (Truong et al., 2023), we perturb ground truth poses with Gaussian noise $\mathcal{N}(0, 0.15)$. For the DTU dataset, we benchmark against BARF, SPARF, BARF + NeuS (Wang et al., 2021a), SPARF + NeuS, and SCNeuS[*1] (Huang et al., 2024c), replacing SuperGlue (Sarlin et al., 2020)

---

[1]We re-implemented it and conducted the evaluation since official code is not available.

|  | scan24 | scan37 | scan65 | scan69 | scan83 | scan105 | scan106 | scan110 | scan118 | scan112 |
|---|---|---|---|---|---|---|---|---|---|---|
| BARF + NeuS | | | | | | | | | | |
| SPARF + NeuS | | | | | | | | | | |
| SCNeuS* | | | | | | | | | | |
| Ours | | | | | | | | | | |
| SparseNeuS (w GT poses) | | | | | | | | | | |

Figure 6: **Qualitative results on the DTU dataset.** We compared A naive adaptation of BARF (Lin et al., 2021) with NeuS (Wang et al., 2021a) and SPARF (Truong et al., 2023) with NeuS (Wang et al., 2021a), SC-NeuS* (Huang et al., 2024c) and Ours. Thanks to occlusion handling and implicit surface smoothing, our method demonstrates superior detail. The last row shows the results of SparseNeuS (Long et al., 2022), which is trained with ground truth camera poses.

Table 1: **Quantitative comparison on the DTU dataset for rotation and translation error.** The upper table shows the rotation errors for each scan, while the lower table shows the translation errors for each scan with randomly augmented noisy camera poses. Both tables indicate the initial errors for each scan at the third row. The best results are in **bold**.

| | SDF | scan | 24 | 37 | 40 | 55 | 63 | 65 | 69 | 83 | 97 | 105 | 106 | 110 | 114 | 118 | 122 | Avg. |
|---|---|---|---|---|---|---|---|---|---|---|---|---|---|---|---|---|---|---|
| | | | | | | | | Rotation(°) (↓) | | | | | | | | | | |
| narrow | | $E_{init}$ | 18.93 | 7.06 | 10.72 | 10.49 | 15.76 | 7.62 | 12.57 | 15.20 | 15.24 | 8.69 | 11.60 | 12.81 | 8.34 | 14.26 | 16.59 | 12.39 |
| narrow | | BARF (Lin et al., 2021) | 3.52 | 0.25 | 7.09 | 11.96 | 0.43 | 0.03 | 3.90 | 4.27 | 18.50 | 5.69 | 10.78 | 3.96 | 7.89 | 11.60 | 6.16 | 6.42 |
| narrow | | SPARF (Truong et al., 2023) | 0.03 | 0.03 | 0.03 | 0.03 | 0.03 | 0.03 | 0.03 | 0.03 | 0.03 | 0.03 | 0.03 | 0.03 | 0.03 | 0.03 | 0.03 | 0.03 |
| narrow | ✓ | BARF (Lin et al., 2021) + NeuS (Wang et al., 2021a) | 20.71 | 4.16 | 10.22 | 12.37 | 11.99 | 9.68 | 3.38 | 12.50 | 13.97 | 12.59 | 10.26 | 14.48 | 0.73 | 12.97 | 7.52 | 10.50 |
| narrow | ✓ | SPARF (Truong et al., 2023) + NeuS (Wang et al., 2021a) | 2.16 | 0.28 | **0.03** | 0.12 | **0.03** | **0.03** | **0.03** | 0.38 | 7.19 | **0.03** | 6.31 | 6.36 | **0.03** | 13.81 | **0.03** | 2.45 |
| narrow | ✓ | SCNeuS* (Huang et al., 2024c) | 0.26 | 3.19 | **0.03** | **0.03** | 2.42 | 0.19 | 0.14 | 0.72 | 0.10 | 0.25 | 0.11 | 0.27 | **0.03** | 0.20 | 0.09 | 2.09 |
| narrow | ✓ | Ours | **0.03** | **0.03** | **0.03** | **0.03** | **0.03** | **0.03** | **0.03** | **0.03** | **0.03** | **0.03** | **0.03** | **0.03** | **0.03** | **0.03** | **0.03** | **0.03** |
| wide | | $E_{init}$ | 18.92 | 8.68 | 10.72 | 10.49 | 15.76 | 7.62 | 12.57 | 12.61 | 13.72 | 8.69 | 18.50 | 12.81 | 7.66 | 14.26 | 18.80 | 12.34 |
| wide | | BARF (Lin et al., 2021) | 16.86 | 6.49 | 11.35 | 8.90 | 8.35 | 7.59 | 13.33 | 13.78 | 10.59 | 21.28 | 11.82 | 9.11 | 17.03 | 11.57 | 12.34 | |
| wide | | SPARF (Truong et al., 2023) | 0.03 | 0.367 | 0.03 | 0.03 | 0.26 | 0.37 | 0.03 | 6.54 | 0.76 | 0.03 | 0.03 | 0.11 | 0.03 | 0.13 | 0.03 | 0.58 |
| wide | ✓ | BARF (Lin et al., 2021) + NeuS (Wang et al., 2021a) | 20.44 | 5.50 | 12.68 | 12.22 | 11.94 | 7.90 | 2.99 | 15.51 | 13.73 | 9.40 | 11.30 | 14.07 | 14.35 | 18.53 | 12.73 | |
| wide | ✓ | SPARF (Truong et al., 2023) + NeuS (Wang et al., 2021a) | 3.37 | **0.03** | **0.03** | **0.03** | 0.56 | 0.18 | **0.03** | 12.92 | 3.51 | **0.03** | 7.75 | 5.96 | 0.17 | 13.35 | **0.03** | 3.20 |
| wide | ✓ | SCNeuS* (Huang et al., 2024c) | **0.03** | 0.50 | **0.03** | **0.03** | **0.03** | 0.04 | 0.08 | 0.14 | 0.11 | 0.26 | 0.06 | 0.25 | 0.04 | 0.13 | 0.10 | 0.12 |
| wide | ✓ | Ours | **0.03** | **0.03** | **0.03** | **0.03** | **0.03** | **0.03** | **0.03** | **0.03** | **0.03** | **0.03** | **0.03** | **0.03** | **0.03** | **0.03** | **0.03** | **0.03** |

| | SDF | scan | 24 | 37 | 40 | 55 | 63 | 65 | 69 | 83 | 97 | 105 | 106 | 110 | 114 | 118 | 122 | Avg. |
|---|---|---|---|---|---|---|---|---|---|---|---|---|---|---|---|---|---|---|
| | | | | | | | | Translation(×100) (↓) | | | | | | | | | | |
| narrow | | $E_{init}$ | 22.90 | 30.89 | 17.00 | 26.78 | 13.40 | 20.24 | 19.26 | 17.41 | 17.47 | 10.72 | 23.50 | 28.42 | 16.65 | 33.26 | 26.83 | 21.65 |
| narrow | | BARF (Lin et al., 2021) | 11.08 | 1.57 | 13.92 | 26.56 | 1.20 | 0.53 | 16.17 | 13.72 | 23.24 | 11.35 | 13.79 | 13.79 | 18.98 | 35.50 | 21.96 | 14.89 |
| narrow | | SPARF (Truong et al., 2023) | 0.26 | 1.01 | 0.46 | 0.34 | 0.95 | 0.48 | 0.16 | 0.48 | 0.15 | 0.33 | 0.22 | 0.84 | 0.16 | 0.2 | 0.13 | 0.41 |
| narrow | ✓ | BARF (Lin et al., 2021) + NeuS (Wang et al., 2021a) | 24.27 | 18.81 | 18.77 | 24.81 | 16.85 | 22.38 | 13.30 | 21.12 | 19.30 | 12.28 | 20.36 | 27.50 | 1.31 | 29.61 | 23.87 | 19.63 |
| narrow | ✓ | SPARF (Truong et al., 2023) + NeuS (Wang et al., 2021a) | 0.31 | 2.80 | 0.25 | 0.42 | 0.61 | 0.47 | 0.26 | 0.73 | 0.33 | 0.20 | 0.22 | 0.43 | **0.24** | **0.20** | **0.11** | 0.51 |
| narrow | ✓ | SCNeuS* (Huang et al., 2024c) | 0.38 | 11.53 | 0.41 | 0.84 | 2.44 | 0.63 | 0.34 | **0.11** | 0.16 | **0.14** | 0.12 | 0.38 | 0.38 | 0.22 | 0.12 | 1.27 |
| narrow | ✓ | Ours | **0.19** | **1.18** | **0.21** | **0.23** | **0.26** | 0.47 | 0.34 | 0.44 | **0.09** | 0.32 | 0.29 | **0.21** | 0.35 | 0.34 | 0.32 | **0.35** |
| wide | | $E_{init}$ | 53.47 | 26.71 | 21.46 | 30.25 | 49.10 | 33.72 | 25.79 | 46.24 | 49.74 | 30.36 | 21.74 | 39.06 | 34.06 | 50.24 | 21.30 | 31.07 |
| wide | | BARF (Lin et al., 2021) | 52.52 | 23.35 | 26.56 | 16.38 | 32.61 | 21.81 | 15.98 | 35.34 | 52.99 | 24.61 | 20.60 | 30.27 | 37.22 | 54.76 | 21.12 | 31.07 |
| wide | | SPARF (Truong et al., 2023) | 0.04 | 0.09 | 0.07 | 0.04 | 0.23 | 0.09 | 0.09 | 2.57 | 0.14 | 0.15 | 0.02 | 0.08 | 0.02 | 0.07 | 0.08 | 0.25 |
| wide | ✓ | BARF (Lin et al., 2021) + NeuS (Wang et al., 2021a) | 53.42 | 23.70 | 17.66 | 28.25 | 47.04 | 23.71 | 26.78 | 50.88 | 49.11 | 33.14 | 36.81 | 37.38 | 32.07 | 56.99 | 23.80 | 36.05 |
| wide | ✓ | SPARF (Truong et al., 2023) + NeuS (Wang et al., 2021a) | 13.41 | **0.98** | 0.69 | 0.39 | 2.44 | 1.36 | 1.15 | 13.02 | 10.37 | 10.37 | 9.52 | 14.27 | 1.78 | 15.28 | 0.60 | 6.37 |
| wide | ✓ | SCNeuS* (Huang et al., 2024c) | 0.97 | 3.43 | 0.78 | 0.33 | **0.60** | **0.12** | **0.21** | 0.67 | 0.20 | 0.85 | 0.15 | 0.42 | **0.13** | 0.32 | 0.33 | 0.63 |
| wide | ✓ | Ours | **0.62** | 1.65 | **0.29** | **0.24** | 1.00 | 0.48 | 0.34 | 0.81 | **0.18** | **0.28** | **0.12** | **0.28** | 0.20 | 0.36 | 0.33 | **0.49** |

with PDCNet (Truong et al., 2021) for fair comparisons. We also include SparseNeuS (Long et al., 2022), MonoSDF (Yu et al., 2022) and Spurfies (Raj et al., 2025), trained with ground truth poses. We skip the results of Spurfies and MonoSDF in qualitative results while the results for SparseNeuS are provided in both qualitative and quantitative comparisons, as it shows best performance among them. For the BlendedMVS dataset, we evaluate SPARF + NeuS and SCNeuS.

## 4.2 RESULTS ON DTU DATASET

**Qualitative results** in the DTU dataset are shown in Fig. 6. BARF struggles to optimize camera poses and build proper 3D reconstructions. SPARF performs better but remains dependent on correspondence matching, causing failures as shown in scan118 in Fig. 6. SCNeuS achieves strong

Table 2: **Quantitative comparison on the DTU dataset for the Chamfer distance.** We report the evaluation results for the Chamfer distance for each scan of the DTU dataset. The top row shows the results of SparseNeuS Long et al. (2022), which was trained with ground truth camera poses without noise. The best results are in **bold**. '-' for the Chamfer distance means that the metric could not be calculated due to absence of any points, resulting from object masking for proper evaluation.

| | | Chamfer Distance (↓) | | | | | | | | | | | | | | | |
|---|---|---|---|---|---|---|---|---|---|---|---|---|---|---|---|---|---|
| | | GT cam | 24 | 37 | 40 | 55 | 63 | 65 | 69 | 83 | 97 | 105 | 106 | 110 | 114 | 118 | 122 | Avg. |
| narrow | BARF (Lin et al., 2021) + NeuS (Wang et al., 2021a) | | 6.03 | 6.11 | 5.27 | 8.69 | - | 7.89 | 7.36 | 5.29 | 6.11 | - | 16.88 | - | 5.61 | - | 7.51 | 7.52 |
| | SPARF (Truong et al., 2023) + NeuS (Wang et al., 2021a) | | 2.11 | 5.33 | **2.18** | 1.35 | 4.52 | 1.72 | **1.19** | 1.75 | 1.87 | **1.11** | 2.01 | 0.84 | 0.78 | **0.99** | **1.37** | 1.94 |
| | SCNeuS* (Huang et al., 2024c) | | 1.92 | 5.92 | 3.41 | 2.63 | 5.48 | 2.94 | 2.10 | 3.90 | 1.58 | 1.86 | 2.21 | 2.10 | **0.59** | 2.48 | 2.38 | 3.11 |
| | Ours | | **1.30** | **3.10** | 2.38 | **1.12** | **1.15** | **1.62** | 1.28 | **1.40** | **1.17** | 1.14 | **1.69** | 0.81 | 0.95 | 1.11 | 1.93 | **1.48** |
| wide | BARF (Lin et al., 2021) + NeuS (Wang et al., 2021a) | | 7.21 | 6.64 | 9.82 | 9.22 | 10.94 | 7.31 | 7.46 | 7.65 | 7.26 | 8.74 | 7.95 | 6.46 | 7.68 | 7.12 | 7.69 | 7.94 |
| | SPARF (Truong et al., 2023) + NeuS (Wang et al., 2021a) | | 6.12 | 4.54 | 3.02 | 0.89 | 4.78 | 5.26 | 1.81 | 8.02 | 2.92 | 2.92 | 7.01 | 6.46 | 0.93 | 6.14 | 2.57 | 4.23 |
| | SCNeuS* (Huang et al., 2024c) | | 3.68 | 4.78 | 4.15 | 0.96 | 3.73 | 5.01 | 1.72 | **2.64** | 1.91 | 2.00 | **1.70** | **1.67** | 0.67 | 1.91 | 2.19 | 2.58 |
| | Ours | | **2.90** | **4.26** | **2.91** | **0.84** | **3.63** | **3.51** | **1.47** | 2.86 | **1.31** | **1.67** | 1.98 | 1.70 | **0.57** | **1.65** | **1.69** | **2.20** |
| | MonoSDF (Yu et al., 2022) | ✓ | 3.47 | 3.61 | 2.10 | 1.05 | 2.37 | 1.37 | 1.41 | 1.85 | 1.74 | 1.10 | 1.46 | 2.28 | 1.25 | 1.44 | 1.45 | 1.86 |
| | Spurfies (Wu et al., 2025) | ✓ | 1.60 | 3.83 | 2.20 | 5.19 | 2.92 | 2.57 | 3.46 | 2.86 | 2.80 | 3.82 | 1.50 | 1.46 | 0.70 | 2.29 | 2.36 | 2.63 |
| | SparseNeuS (Long et al., 2022) | ✓ | 1.29 | 2.27 | 1.57 | 0.88 | 1.61 | 1.86 | 1.06 | 1.27 | 1.42 | 1.07 | 0.99 | 0.87 | 0.54 | 1.15 | 1.18 | 1.27 |

results due to the photometric consistency loss but struggles with occlusions in large baseline scenarios. Our method outperforms previous approaches, effectively handling occlusion in large-baseline scenarios and leveraging surface smoothing for stable surface learning, performing comparably to SparseNeuS, which benefits from ground truth poses.

**Quantitative results** in DTU dataset are provided in Table 1 and Table 2 for camera pose errors and the Chamfer distance, respectively. For camera pose estimation, SPARF maintains stability in the large baselines but suffers when incorporating implicit surface priors. SCNeuS is quantitatively strong but degrades with increasing camera baseline. Our approach consistently achieves lower rotation and translation errors than previous works. Note that the lowest rotation error $0.03°$ is limited by the alignment function, which includes an eps $1e^{-6}$ to prevent numerical instability. The detail would be introduced in Appendix B.3. For the Chamfer distance, SPARF + NeuS outperforms SCNeuS in the narrow baseline, but as the baseline widen, SCNeuS benefits from patch-wise NCC loss. Our method, incorporating enhanced view consistency constraints and surface smoothing, delivers robust reconstructions with superior performance.

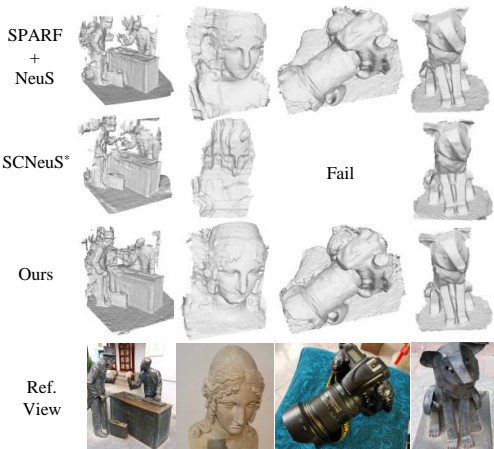

Figure 7: **Qualitative results on BlendedMVS dataset** with randomly chosen 4 images. From the left, we denote each scene as A, B, C and D for convenience, respectively.

### 4.3 RESULTS ON BLENDEDMVS DATASET

**Qualitative results** in BlendedMVS dataset are shown in Fig. 7. We evaluated 4 randomly chosen scenes, denoted as A, B, C, and D, with their exact names provided in the Appendix B.5, using SPARF+NeuS, SCNeuS and our method. Our method demonstrates robustness in high-frequency regions, where others show low performance. Especially SCNeuS, it fails to reconstruct proper 3D surfaces at scene C due to large camera error during optimization.

Table 3: **Quantitative comparison on the BlendedMVS dataset.** For most selected scenarios, our method shows robust performance on the camera pose refinement.

| metric | Rotation(°) (↓) | | | | | Translation(× 100) (↓) | | | | |
|---|---|---|---|---|---|---|---|---|---|---|
| scene | A | B | C | D | Avg. | A | B | C | D | Avg. |
| $E_{init}$ | 11.88 | 11.86 | 12.47 | 12.20 | 12.10 | 9.03 | 24.57 | 40.84 | 42.93 | 29.34 |
| SPARF+NeuS | **0.03** | 1.16 | **0.03** | 0.73 | 0.49 | **0.04** | 0.90 | 0.66 | 1.71 | 0.83 |
| SCNeuS* | **0.03** | 16.17 | 1.89 | 7.45 | 6.385 | 0.20 | 37.57 | 4.63 | 37.36 | 19.94 |
| Ours | **0.03** | **0.03** | **0.03** | **0.03** | **0.03** | 0.06 | **0.39** | **0.41** | **0.46** | **0.33** |

**Quantitative results** in BlendedMVS dataset are shown in Table 3. In general, SPARF + NeuS demonstrates better performance than SCNeuS, even without a specialized handling of occlusions. Our method maintains stable performance across various sparse-view scenarios, achieving superior camera pose estimation, while other approaches struggle with correcting camera poses in the large baseline.

## 4.4 Ablation Study

In this paper, we introduced two simple techniques for handling sparse and noisy input views: uncertainty-aware phometric/geometric consistency and implicit surface smoothing. In this section, we demonstrate the effectiveness of our approach by analyzing the role of uncertainty in refining noisy camera poses and the impact of implicit surface smoothing on visual performance.

Table 4: Ablation study of uncertainty and view-consistency losses. Initial errors for rotation and translation are reported in bottom row. The best results are in **bold**.

| Exp. | Preserve (✓) | | | Evaluation | |
|---|---|---|---|---|---|
| | Uncertainty | Patch-wise NCC | Geometric Consistency | R↓ | t↓ |
| A | | | | 5.96 | 14.27 |
| B | | ✓ | | 0.25 | 0.42 |
| C | | | ✓ | 7.57 | 13.55 |
| D | | ✓ | ✓ | **0.03** | 0.39 |
| E | ✓ | ✓ | | **0.03** | 0.29 |
| F | ✓ | | ✓ | **0.03** | 0.57 |
| G | ✓ | ✓ | ✓ | **0.03** | **0.28** |
| | | | $E_{init}$ | 12.81 | 39.06 |

**Uncertainty** is designed by the normalized discrepancy of the 3D coordinates of correspondences. We present an ablation study for the proposed technique in Table 4, conducted on DTU scan110, to illustrate the impact of uncertainty on occlusion-aware NCC loss and geometric consistency. The results show that the proposed algorithms gradually improve performance as each component is added. Note that experiment A of Table 4 corresponds to SPARF (Truong et al., 2023) with NeuS (Wang et al., 2021a), B represents SCNeuS (Huang et al., 2024c), and G represents our method. As reported in SCNeuS (Huang et al., 2024c), patch-wise NCC loss plays a key role in refining camera pose, and also, we observe that the geometric consistency loss also demonstrates improvement when combined with uncertainty. For other scenes, we provide full ablation studies in the Appendix C.2.

**Implicit surface smoothing**, implemented using the pSDF loss, is compared to a version with only the Eikonal loss, as shown in Fig. 8. While Eikonal loss struggles to create reliable surfaces, our method produces clear surfaces, highlighting the effectiveness of implicit surface smoothing. This enables successful surface reconstruction even with inaccurately aligned camera poses, as shown by the quantitative results in Table 2. Due to space limitations, we have included the quantitative results for other DTU dataset scenes in Appendix C.1, which further describe the effectiveness of the pSDF loss.

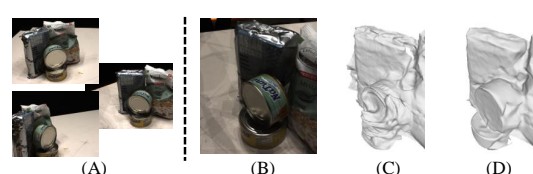

(A)   (B)   (C)   (D)

Figure 8: Visualization of the ablation study for surface smoothing loss. From left to right: (A) training input views, (B) reference view, (C) result without surface smoothing loss, and (D) ours. Note that (C) and (D) are results from the same camera positions.

## 5 Conclusion

In this paper, we present an effective approach for reconstructing implicit surfaces from sparse and noisy views. Unlike other methods that overlook occlusion handling or optimize surfaces under uncertain camera poses, we explore techniques for optimizing models in challenging environments. Our contributions include an occlusion-handling approach with view-consistency losses and an implicit surface smoothing technique, enabling the model to learn surface geometries effectively even under imperfect conditions. As a result, our method achieves state-of-the-art performance in both refining camera poses and surface reconstruction. However, limitations remain. Although patch-wise NCC loss aids in optimizing noisy camera poses, our approach still heavily relies on matching correspondences. Incorporating the proposed techniques with strong dense matching works, such as DUSt3R (Wang et al., 2024), VGGT (Wang et al., 2025a), $\pi^3$ (Wang et al., 2025b), may help to overcome an upperbound. Furthermore, sparse guidance for dense geometry can lead to unwanted floating surfaces, known as *floaters*. Although methods like FreeNeRF (Yang et al., 2023) and SparseNeuS (Long et al., 2022) introduced additional regularization techniques, they still face challenges in preventing floaters in harsh scenarios. This issue may be mitigated by incorporating geometric supervision from off-the-shelf models, as Depth-Pro (Bochkovskii et al., 2025).

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

# A APPENDIX

In the appendix, we provide additional descriptions of the implementation details, the terms used in the progressive SDF (pSDF) loss, the evaluation protocols, detailed information of experiments, and the relationship between PDCNet (Truong et al., 2021) which can also predict a confidence and uncertainty that we proposed. In addition, it includes additional ablation studies on the uncertainty and view-consistency losses.

# B IMPLEMENTATION DETAILS

We provide further information about the implementation in this section, including the architecture, the components of the pSDF loss, the evaluation protocols, the names of scenes tested in the BlendedMVS dataset, and a comparison between the confidence map predicted by PDCNet and the uncertainty.

## B.1 ARCHITECTURE

We used baseline models for SPARF (Truong et al., 2023) and NeuS (Wang et al., 2021a). For details of the architecture, we follow the same configuration as in NeuS (Wang et al., 2021a). The learning rate is initialized to $10^{-3}$ and decays to $10^{-4}$ for an Adam optimizer with an ExponentialLR scheduler, which is utilized for both the model and the camera optimizers. To optimize the pose, we employ the same parameterization strategy as in SPARF (Truong et al., 2023). The two-column method is used for all experiments, including SCNeuS (Huang et al., 2024c), following SPARF (Truong et al., 2023). For the weight factors $\lambda$ used in the loss functions described in the manuscript, we set $[\lambda_{M.C.}, \lambda_{u,gc}, \lambda_{pSDF}, \lambda_{occ}, \lambda_{geo}] = [10^{-3}, 0.1, 0.1, 0.01, 10^{-3}]$. All experiments are conducted with 100k iterations.

## B.2 PSDF LOSS

We proposed the progressive SDF loss, to smooth surfaces. The function $h(k, b(k))$ of the manuscript defines the smoothing area. The threshold value $b(k)$ decreases as the iteration k increases, such that:

$$b(k) = \lambda_b cos \frac{A(k)\pi}{2} d \tag{12}$$

where $A(k) = \frac{k-\alpha_1}{\alpha_2-\alpha_1}$ and $k \in [\alpha_1, \alpha_2]$. Here, $\alpha_1$ is the starting point and $\alpha_2$ is the end point for the coarse-to-fine strategy, following BARF (Lin et al., 2021) and SPARF (Truong et al., 2023). We observed that when $b(k)$ has a large value, the large area is affected, potentially creating planar surfaces and destabilizing performance. To address this, we introduce a scaling factor $\lambda_b$ to reduce the bound $b(k)$. We set $\lambda_d = 0.1$.

With these parameters, the smoothing area function $h(k, b(k))$ is defined as:

$$h(k, b(k)) = \begin{cases} 0 & \text{if} \quad k < \alpha_1 \\ b(k) & \text{if} \quad \alpha_1 \leq k < \alpha_2 \\ 0 & \text{if} \quad k \geq \alpha_2. \end{cases} \tag{13}$$

## B.3 EVALUATION FOR ROTATION ERROR

To evaluate camera poses for rotation error and translation error, we used the same strategy of SPARF (Truong et al., 2023). The detailed process is in Alg. 1. Note that the rotation error 0.03 is caused by an eps value $\epsilon$ to avoid the numerical instability of the $\arccos(\cdot)$ operation.

---

**Algorithm 1:** Camera Pose Evaluation

---

**Input** : Estimated world-to-camera poses $P_{est}^{w2c} \in \mathbb{R}^{B \times 4 \times 4}$, Ground-truth world-to-camera poses $P_{GT}^{w2c} \in \mathbb{R}^{B \times 4 \times 4}$

**Output:** Rotation error $\mathrm{E}_R$, Translation error $\mathrm{E}_t$, scale $s$

$P_{est}^{c2w} \leftarrow \mathrm{invert}(P_{est}^{w2c})$
$P_{GT}^{c2w} \leftarrow \mathrm{invert}(P_{GT}^{w2c})$

// Split into rotation and translation
$\mathbf{R}_{est}^{c2w}, \mathbf{t}_{est}^{c2w} \leftarrow \mathrm{split}(P_{est}^{c2w}[:,:3,:], \mathrm{dims} = [3,1], \mathrm{axis} = -1)$
$\mathbf{R}_{GT}^{c2w}, \mathbf{t}_{GT}^{c2w} \leftarrow \mathrm{split}(P_{GT}^{c2w}[:,:3,:], \mathrm{dims} = [3,1], \mathrm{axis} = -1)$

// Iterate through pose pairs for robust alignment
best_error $\leftarrow \infty$
best_$\mathrm{E}_R \leftarrow \infty$
best_$\mathrm{E}_t \leftarrow \infty$
$\mathbf{P}_{best} \leftarrow \emptyset$
**for** *all pairs* $(id_a, id_b)$ *where* $id_a, id_b \in \{0, \ldots, \min(B, 9)\}, id_a \neq id_b$ **do**
 // Find the transformation
 $dist_{est} \leftarrow \mathrm{norm}(\mathbf{t}_{est}^{c2w}[id_a] - \mathbf{t}_{est}^{c2w}[id_b])$
 $dist_{GT} \leftarrow \mathrm{norm}(\mathbf{t}_{GT}^{c2w}[id_a] - \mathbf{t}_{GT}^{c2w}[id_b])$
 $s \leftarrow dist_{GT}/dist_{est}$
 $P_{est}^{c2w}[:,:3,3] \leftarrow P_{est}^{c2w}[:,:3,3] \times s$      // Apply scale
 $\mathbf{T} \leftarrow P_{GT}^{c2w}[id_a] \times \mathrm{invert}(P_{est}^{c2w}[id_a])$
 $P_{temp-aligned}^{c2w} \leftarrow \mathbf{T} \times P_{est}^{c2w}$
 $P_{pair\_aligned}^{w2c} \leftarrow \mathrm{invert}(P_{temp-aligned}^{c2w})$
 // Evaluate error
 error, $\mathrm{E}_R, \mathrm{E}_t \leftarrow \mathrm{evaluate}(P_{pair\_aligned}^{w2c}, P_{GT}^{w2c})$
 // Select the best alignment
 **if** *error* < *best_error* **then**
  best_error $\leftarrow$ error
  best_$\mathrm{E}_R \leftarrow \mathrm{E}_R$
  best_$\mathrm{E}_t \leftarrow \mathrm{E}_t$
  $P_{best} \leftarrow P_{pair\_aligned}^{w2c}$

// Return best values
$P_{aligned}^{c2w} \leftarrow \mathrm{invert}(P_{best})$
$\mathrm{E}_R \leftarrow$ best_$\mathrm{E}_R$
$\mathrm{E}_t \leftarrow$ best_$\mathrm{E}_t$

---

---

**Algorithm 2:** Evaluation Function for Camera Poses

---

**Input** : world-to-camera poses $P^{w2c} \in \mathbb{R}^{B \times 4 \times 4}$, target world-to-camera poses $P_{tar}^{w2c} \in \mathbb{R}^{B \times 4 \times 4}$

**Output:** total_error, Rotation error $\mathrm{E}_R$, Translation error $\mathrm{E}_t$

// Evaluate rotation error
$\epsilon \leftarrow 1 \times 10^{-6}$           // for numerical stability
$\mathbf{R}^{c2w}, \mathbf{t}^{c2w} \leftarrow \mathrm{split}(P^{c2w}[:,:3,:], \mathrm{dims} = [3,1], \mathrm{axis} = -1)$
$\mathbf{R}_{tar}^{c2w}, \mathbf{t}_{tar}^{c2w} \leftarrow \mathrm{split}(P_{tar}^{c2w}[:,:3,:], \mathrm{dims} = [3,1], \mathrm{axis} = -1)$
$\mathbf{R}_{diff} \leftarrow \mathbf{R}^{c2w} \times (\mathbf{R}_{tar}^{c2w})^T$
$trace \leftarrow \mathrm{tr}(\mathbf{R}_{diff})$
$\mathbf{E}_R \leftarrow \arccos(\mathrm{clamp}((trace - 1)/2, -1 + \epsilon, 1 - \epsilon))$
// Evaluate translation error
$\mathbf{E}_t \leftarrow \mathrm{norm}(\mathbf{t}^{c2w} - \mathbf{t}_{tar}^{c2w})$
total_error = $\mathbf{E}_R \times \mathbf{E}_t$

---

### B.4 Chamfer Distance Evaluation

To compute the Chamfer distance, we align the predicted point cloud set $\mathcal{PC}$ with the ground-truth point cloud set $\hat{\mathcal{PC}}$ using Open3D (Zhou et al., 2018) point cloud registration methods. Following standard guidelines, we first perform RANSAC to obtain an initial global registration. Next, we apply point-to-point Iterative Closest Point (ICP) algorithm for local registration in all experiments. We heuristically found that applying a prealignment transformation using a scale factor and aligning matrices, which were computed during camera pose evaluation, improved the precision of the evaluation.

---

**Algorithm 3:** Point Cloud Registration and Alignment

---

**Input** : Source vertices $\mathbf{V}_s \in \mathbb{R}^3$, aligning camera matrices $\mathbf{R} \in \mathbb{R}^{3\times3}$ and $\mathbf{t} \in \mathbb{R}^3$, scale factor $s$, ground truth point cloud $\hat{\mathcal{PC}}$, threshold value for registration $\text{thr}_{icp}$.

**Function:** In Open3D,
    `PointCloud`: Converts vertices to a point cloud object.
    `PointCloud.transform`: Applies a rigid body transformation.
    `execute_global_registration`: Performs global alignment with RANSAC.
    `registration_icp`: Refines alignment matrices using the ICP algorithm.

**Output** : Aligned point cloud $\mathbf{V}_{aligned}$

---

```
// Prealign from source coordinate to the GT coordinate
```
$\mathbf{V}_{scaled} \leftarrow s(\mathbf{R} \times \mathbf{V}_s + \mathbf{t})$
```
// Create point cloud objects
```
$\mathcal{PC} \leftarrow \text{PointCloud}(\mathbf{V}_{scaled})$
```
// Perform global registration
```
$\text{result}_{ransac} \leftarrow \text{execute\_global\_registration}(\mathcal{PC}, \hat{\mathcal{PC}})$
```
// Refine registration using a local ICP algorithm
```
$\text{reg}_{p2l} \leftarrow \text{registration\_icp}(\mathcal{PC}, \hat{\mathcal{PC}}, \text{thr}_{icp}, \text{result}_{ransac}.)$
```
// Apply the refined transformation
```
$\mathbf{V}_{algined} \leftarrow \text{asarray}(\mathcal{PC}.\text{transform}(\text{reg}_{p2l}.\text{transformation}).\text{points})$

---

### B.5 Used Scene in BlendedMVS Dataset

Scenes A, B, C and D, used in the BlendedMVS dataset (Yao et al., 2020) in the main manuscript, are: 58cf4771d0f5fb221defe6da, 59f363a8b45be22330016cad, 5c34300a73a8df509add216d and 5c1af2e2bee9a7423c963d019, respectively.

### B.6 PDCNet and Uncertainty

It should be noted that the confidence map generated by PDCNet (Truong et al., 2021) could potentially be inverted to function as an uncertainty map. However, its utility for our method is limited by the fact that PDCNet provides confidence scores for its predicted matching points, rather than for specific correspondences within an image pair or arbitrary spatial locations. For example, if $x_2$ is predicted to match $y_2$ by PDCNet, it does not provide a confidence score for $(x_2, y_1)$, where $y_1 \neq y_2$. Since the sampled rays are random, recalculating confidence scores for arbitrary points would require an additional neural network. Therefore, we did not employ the confidence map of PDCNet for uncertainty, as our approach requires a method that can be derived from arbitrary points $(\mathbf{p}, \mathbf{q})$ which are sampled from the camera poses being optimized during training.

Table 5: Additional ablation study of the pSDF loss. The best results are in **bold**. The pSDF loss is only applied to *Ours*.

| Exp. | pSDF | Chamfer Distance($\downarrow$) | | | | | | | | | | | | | | | |
|---|---|---|---|---|---|---|---|---|---|---|---|---|---|---|---|---|---|
| | | 24 | 37 | 40 | 55 | 63 | 65 | 69 | 83 | 97 | 105 | 106 | 110 | 114 | 118 | 122 | Avg. |
| Ours w/o pSDF | | 4.11 | 5.51 | 3.58 | 0.92 | 4.71 | 4.71 | **1.32** | 3.07 | 2.11 | **1.55** | 1.99 | **1.37** | 0.67 | 1.93 | **1.61** | 2.61 |
| Ours | ✓ | **2.90** | **4.26** | **2.91** | **0.84** | **3.63** | **3.51** | 1.47 | **2.86** | **1.31** | 1.67 | **1.98** | 1.70 | **0.57** | **1.65** | 1.69 | **2.20** |

Table 6: Ablation study of uncertainty and view-consistency losses. The upper table shows the rotation error and the lower table shows the translation error. U. means uncertainty, G.C. means geometric consistency. The best results are in **bold**.

| | Rotation(°) (↓) | | | | | | | | | | | | | | | | | | |
|---|---|---|---|---|---|---|---|---|---|---|---|---|---|---|---|---|---|---|---|
| Exp. | Preserve (✓) | | | 24 | 37 | 40 | 55 | 63 | 65 | 69 | 83 | 97 | 105 | 106 | 110 | 114 | 118 | 122 | Avg. |
| | U. | NCC | G.C. | | | | | | | | | | | | | | | | |
| A | | | | 3.37 | **0.03** | **0.03** | **0.03** | 0.56 | 0.18 | **0.03** | 12.92 | 3.51 | **0.03** | 7.75 | 5.98 | 0.17 | 13.35 | **0.03** | 3.20 |
| B | | ✓ | | **0.03** | 0.50 | **0.03** | **0.03** | **0.03** | 0.04 | 0.08 | 0.14 | 0.11 | 0.26 | 0.06 | 0.25 | 0.04 | 0.13 | 0.10 | 0.12 |
| C | | | ✓ | **0.03** | 0.18 | **0.03** | **0.03** | **0.03** | **0.03** | **0.03** | 0.86 | 1.81 | **0.03** | 4.96 | 7.57 | **0.03** | **0.03** | **0.03** | 1.05 |
| D | | ✓ | ✓ | **0.03** | **0.03** | **0.03** | **0.03** | **0.03** | **0.03** | **0.03** | **0.03** | **0.03** | **0.03** | **0.03** | **0.03** | **0.03** | **0.03** | **0.03** | **0.03** |
| E | ✓ | ✓ | | **0.03** | 0.09 | **0.03** | **0.03** | 0.07 | **0.03** | 0.20 | 0.49 | 6.91 | **0.03** | **0.03** | **0.03** | **0.03** | 0.29 | **0.03** | 0.55 |
| F | ✓ | | ✓ | **0.03** | **0.03** | **0.03** | **0.03** | **0.03** | 4.34 | **0.03** | 0.14 | 10.20 | **0.03** | 11.67 | **0.03** | **0.03** | **0.03** | **0.03** | 1.78 |
| G | ✓ | ✓ | ✓ | **0.03** | **0.03** | **0.03** | **0.03** | **0.03** | **0.03** | **0.03** | **0.03** | **0.03** | **0.03** | **0.03** | **0.03** | **0.03** | **0.03** | **0.03** | **0.03** |

| | Translation (↓) | | | | | | | | | | | | | | | | | | |
|---|---|---|---|---|---|---|---|---|---|---|---|---|---|---|---|---|---|---|---|
| Exp. | Preserve (✓) | | | 24 | 37 | 40 | 55 | 63 | 65 | 69 | 83 | 97 | 105 | 106 | 110 | 114 | 118 | 122 | Avg. |
| | U. | NCC | G.C. | | | | | | | | | | | | | | | | |
| A | | | | 13.41 | **0.98** | 0.69 | 0.39 | 2.44 | 1.36 | 1.15 | 13.02 | 10.37 | 10.36 | 9.52 | 14.27 | 1.78 | 15.28 | 0.60 | 6.37 |
| B | | ✓ | | 0.97 | 3.43 | 0.78 | 0.33 | 0.60 | **0.12** | **0.21** | 0.67 | 0.20 | 0.85 | **0.15** | 0.42 | 0.13 | 0.33 | **0.33** | 0.63 |
| C | | | ✓ | 0.39 | 2.32 | 0.80 | 0.38 | 0.87 | 0.75 | 0.75 | 4.22 | 6.13 | 0.63 | 8.80 | 13.55 | 0.15 | 0.35 | 0.96 | 2.74 |
| D | | ✓ | ✓ | 0.51 | 1.30 | 1.12 | 0.29 | **0.55** | 0.61 | 0.70 | **0.49** | 0.40 | 0.54 | 0.63 | 0.39 | **0.11** | 0.65 | 0.75 | 0.60 |
| E | ✓ | ✓ | | **0.25** | 1.89 | 0.47 | 0.28 | 1.96 | 1.25 | 1.20 | 2.57 | 16.73 | 0.87 | 0.76 | 0.29 | 0.20 | 1.29 | 0.24 | 2.02 |
| F | ✓ | | ✓ | 0.63 | 1.42 | 1.19 | 0.57 | 1.07 | 5.69 | 1.16 | 1.99 | 23.21 | 0.63 | 18.71 | 0.57 | 0.17 | **0.27** | 0.91 | 3.88 |
| G | ✓ | ✓ | ✓ | 0.62 | 1.65 | **0.29** | **0.24** | 1.00 | 0.48 | 0.34 | 0.81 | **0.18** | **0.28** | 0.27 | **0.28** | 0.20 | 0.36 | **0.33** | **0.49** |

# C   ADDITIONAL EXPERIMENTS

In this section, we present two additional analyses that contain ablation studies for pSDF loss by measuring the Chamfer distance and for view-consistency losses and uncertainty to validate the effectiveness of them with camera pose error.

## C.1   ABLATION STUDY FOR PSDF LOSS

In the main manuscript, we present only the qualitative results of the ablation study for the pSDF loss due to space limitation. To enhance understanding and support our results, we present quantitative results of the ablation study for the pSDF loss in the full scans of the DTU dataset as shown in Table 5. Ours shows better performance.

## C.2   ABLATION STUDY FOR CONSISTENCY LOSSES

In this section, we provide additional analyses for view-consistency losses and uncertainty. In the manuscript, the Table 4 only presents the performance of the scan110 of the DTU dataset. To further enhance understanding, we present full evaluation results on the scans of the DTU dataset in Table 6.

## C.3   COMPARISON WITH FEED-FORWARD APPROACHES

Recent feed-forward dense matching algorithms have shown strong performance in 3D reconstruction, especially in the sparse-view setting. Due to these impressive results, many recent works, including DUSt3R (Wang et al., 2024), have adopted feed-forward pipelines. However, our method primarily focuses on refining noisy camera poses rather than predicting them for initialization, and is therefore more appropriately positioned as a post-processing step. We argue that feed-forward methods often require additional optimization stages (e.g. InstantSplat (Fan et al., 2024)) to achieve higher accuracy. Our approach, as well as the baselines compared in the manuscript, can be categorized as optimization-based techniques that aim to refine camera poses and improve surface reconstruction quality.

To support this argument, we performed a simple experiment using InstantSplat (Müller et al., 2022) + 2DGS (Huang et al., 2024a) initialized with MASt3R (Leroy et al., 2024). All experiments were performed on two input images (000022.png and 000028.png) from the scan24 of the DTU dataset. The results are presented in Table 7. As shown, InstantSplat, even with its camera pose refinement, performs worse than the initialization of MASt3R in camera pose evaluation. Our method shows better performance in rotation error, leading to an improved Chamfer distance.

Table 7: Ablation study of uncertainty and view-consistency losses in gaussian splatting applications. The best results are in **bold**.

| Exp. | Evaluation | | |
|---|---|---|---|
| | R$\downarrow$ | t$\downarrow$ | CD$\downarrow$ |
| MASt3R (Leroy et al., 2024) | 0.52 | **1.12** | 2.33 |
| InstantSplat (Fan et al., 2024) + 2DGS (Huang et al., 2024a) | 0.54 | **1.12** | 2.28 |
| InstantSplat (Fan et al., 2024) + 2DGS (Huang et al., 2024a) + Ours | **0.45** | **1.12** | **2.24** |

Note that the progressive SDF loss was not implemented in this experiment, as it is designed for ray-based methods with SDF and is not directly applicable to 2D Gaussian Splatting (Huang et al., 2024a).

