# OpenReview forum: "Implicit Surface Reconstruction from Sparse and Noisy Poses with Large Motions"
_ICLR.cc/2026/Conference — Submitted to ICLR 2026_

### Official Review · Reviewer_SsPa · 2025-10-31

**Soundness:** 2
**Presentation:** 3
**Contribution:** 2
**Rating:** 4
**Confidence:** 4

**Summary:**

This paper improves neural implicit surface reconstruction from sparse and noisy poses by using uncertainty-aware guided multi-view consistency. The method models the uncertainty by computing the unprojected errors of correspondences based on rendered depths. Then, the method incorporates this uncertainty into patch-wise NCC and  unprojection errors. In addition, the method introduces progressive SDF loss to smooth implicit surfaces. The experiments show that the method can achieve competitive performance on DTU under narrow and wide baselines.

**Strengths:**

1. The method leverages uncertainty to focus on high confident photometric and geometric information to refine noisy poses.
2. The method follows iSDF and uses coarse geoemtry in early training to impose progressive SDF loss.
3. The method is simple and easy to follow.

**Weaknesses:**

1. The ablation study is only conducted on one sene, DTU 110 scan. This experiment is not convincing.
2. In fact, the method uses patch-wise NCC to help surface reconstruction. Then, the baseline used in this paper, SPARF + NeuS is unfair as NeuS does not use patch-wise NCC. A fair baseline should be SPARF + Geo-NeuS.
3. In ablation study, it seems that the better sufaces with progressive SDF loss cannot help pose refinement.

**Questions:**

1. To better validate the effiectiveness of the introduced strategies, it is better to conduct ablation study on all DTU scans.
2. As SPARF can provide competitive or better pose estimation and the method uses patch-wise NCC, the baseline should be repalced with SPARF + Geo-NeuS, which combines NeuS with patch-wise NCC to verify the surface reconstruction performance.
3. With progressive SDF loss, the surface reconstruction becomes better, I wonder if this can help pose refinement.
4. For uncertainty modeling, there exist many NeRF works model uncertainty with different stragtegies, such as ActiveNeRF [1] and AR-NeRF [2]. I wonder if these strategies can replace the uncertainty modeling in this work.
[1] Pan et al. ActiveNeRF: Learning where to See with Uncertainty Estimation, ECCV 2022.
[2] Xu et al. Few-shot NeRF by Adaptive Rendering Loss Regularization. ECCV 2024.

---

> ### Author Response · Authors · 2025-11-13
>
> Thank you for the precious comments by reviewer SsPa. Your observations are critical for strengthening the clarity and scope of our paper.
>
> - The complete quantitative ablation results across all DTU scans are presented in the appendix: Table 5 (Appendix C.1) for the Chamfer distance and Table 6 (Appendix C.2) for rotation and translation errors.
> - We agree with the reviewer's suggestion. We will add SPARF + Geo-NeuS [1] as an additional baseline, and we will report the results as a comment (ETA: Nov. 25th).
>
> - pSDF primarily stabilizes geometry under noisy poses. As indicated in Lines 459–461, applying the pSDF loss before establishing the initial coarse geometry can cause over-smoothing and performance degradation, often tending toward planar surfaces. Therefore, we intentionally apply the pSDF loss only after the initial coarse geometry is established, a stage which coincides with when camera pose updates are ceased. We recognize this constraint was not explicitly explained in the implementation details. We thank the reviewer for this precise observation.
>
> - Our uncertainty is deterministic and render-aware 3D. Methods like ActiveNeRF [2] and AR-NeRF [3] estimate uncertainty, an approach orthogonal to ours. We will perform an additional ablation study replacing our uncertainty $u$ with the predicted uncertainty from ActiveNeRF. Since the code for AR-NeRF is unavailable, we will focus on the comparison with the ActiveNeRF [2] methodology. The experiment requires more time, and we will update the results in a subsequent comment. (ETA: December 1st)
>
> [1] Fu, Qiancheng, et al. "Geo-neus: Geometry-consistent neural implicit surfaces learning for multi-view reconstruction." Advances in Neural Information Processing Systems 35 (2022): 3403-3416.
>
> [2] Pan et al. ActiveNeRF: Learning where to See with Uncertainty Estimation, ECCV 2022.
>
> [3] Xu et al. Few-shot NeRF by Adaptive Rendering Loss Regularization. ECCV 2024.

---

> > ### Author Response · Authors · 2025-11-28
> >
> > We provide the results from replacing our uncertainty with the ActiveNeRF methodology. For fair comparison, all experiments utilize the patch-wise NCC loss and geometric consistency loss. Since PDCNet is not designed for arbitrary sample uncertainty, we excluded it from this experiment. This matches the setting of Table 4 (Main Manuscript) and Table 6 (Appendix). The results are as follows:
> >
> > For the rotation error:
> > |           |   24  |   37  |   40  |   55  |   63  |  65  |   69  |   83  |   97  |  105  |  106  |  110  |  114 |  118  |  122  |  Avg  |
> > |:----------:|:-----:|:-----:|:-----:|:-----:|:-----:|:----:|:-----:|:-----:|:-----:|:-----:|:-----:|:-----:|:----:|:-----:|:-----:|:-----:|
> > |   E_init   | 20.01 | 11.87 | 12.83 | 11.11 | 17.67 | 7.94 | 19.63 | 13.09 | 19.44 | 12.83 | 14.34 | 16.48 | 8.57 | 16.16 | 20.92 | 14.86 |
> > |   -   |  3.37 |  0.03 |  0.03 |  0.03 |  0.56 | 0.18 |  0.03 | 12.92 |  3.51 |  0.03 |  7.75 |  5.98 | 0.17 | 13.35 |  0.03 |  3.20 |
> > | ActiveNeRF |  1.74 |  0.38 |  0.03 |  0.03 |  1.38 | 0.03 |  0.03 |  0.53 |  3.44 |  0.03 |  3.96 |  3.96 | 0.03 |  7.26 |  0.03 |  1.52 |
> > |    Ours    |  0.03 |  0.03 |  0.03 |  0.03 |  0.03 | 0.03 |  0.03 |  0.03 |  0.03 |  0.03 |  0.03 |  0.03 | 0.03 |  0.03 |  0.03 |  0.03 |
> >
> > For the translation error:
> > |           |   24  |   37  |   40  |   55  |   63  |   65  |   69  |   83  |   97  |  105  |  106  |  110  |  114  |  118  |  122  |  Avg  |
> > |:----------:|:-----:|:-----:|:-----:|:-----:|:-----:|:-----:|:-----:|:-----:|:-----:|:-----:|:-----:|:-----:|:-----:|:-----:|:-----:|:-----:|
> > |   E_init   | 71.83 | 42.39 | 34.41 | 51.35 | 67.75 | 45.91 | 49.16 | 48.15 | 65.69 | 61.87 | 31.84 | 53.03 | 30.48 | 52.92 | 41.04 | 49.86 |
> > |  -  |  0.51 |  1.30 |  1.12 |  0.29 |  0.55 |  0.61 |  0.70 |  0.49 |  0.40 |  0.54 |  0.63 |  0.39 |  0.11 |  0.65 |  0.75 |  0.60 |
> > | ActiveNeRF |  5.51 |  4.88 |  0.54 |  0.48 |  3.07 |  0.68 |  0.37 |  3.95 | 10.43 |  0.70 |  0.98 |  9.30 |  0.52 |  7.35 |  0.26 |  3.27 |
> > |    Ours    |  0.62 |  1.65 |  0.29 |  0.24 |  1.00 |  0.48 |  0.34 |  0.81 |  0.18 |  0.28 |  0.27 |  0.28 |  0.20 |  0.36 |  0.33 |  0.49 |
> >
> > For the Chamfer Distance:
> > |           |  24  |  37  |  40  |  55  |  63  |  65  |  69  |  83  |  97  |  105 |  106 |  110 |  114 |  118 |  122 |  Avg |
> > |:----------:|:----:|:----:|:----:|:----:|:----:|:----:|:----:|:----:|:----:|:----:|:----:|:----:|:----:|:----:|:----:|:----:|
> > |  -  | 3.78 | 4.58 | 3.64 | 1.54 | 3.15 | 5.64 | 1.29 | 2.23 | 1.67 | 1.45 | 3.77 |   -   | 0.57 | 2.05 | 1.51 | 2.63 |
> > | ActiveNeRF | 5.00 | 5.21 | 3.07 | 1.59 | 5.43 | 3.47 | 2.14 | 0.03 | 2.99 | 1.64 | 5.30 | 4.70 | 0.94 | 6.85 | 1.85 | 3.35 |
> > |    Ours    | 2.90 | 4.26 | 2.91 | 0.84 | 3.63 | 3.51 | 1.47 | 2.86 | 1.31 | 1.67 | 1.98 | 1.70 | 0.57 | 1.65 | 1.69 | 2.20 |
> >
> > The results demonstrate that ActiveNeRF performs worse than both our method and the baseline without any uncertainty weighting (marked '-'), particularly in translation. This degradation likely occurs because ActiveNeRF's implicit uncertainty struggles to collect reliable information in sparse, noisy settings. Consequently, the low or inaccurate uncertainty estimates lead to incorrect weighting of the losses, resulting in degraded final performance. This highlights that for wide-baseline scenarios, our geometry-derived uncertainty is superior to the network-predicted uncertainty.

---

> ### Author Response · Authors · 2025-11-24
>
> We performed additional experiment, Geo-NeuS [1]. The experiments were conducted on the DTU large-baseline setup, matching the scenario presented in Table 1 of the main manuscript. The results are as follows:
>
> For the rotation error:
> |                   |    24 |   37 |    40 |    55 |    63 |    65 |    69 |    83 |    97 |  105 |   106 |   110 |   114 |   118 |   122 | Avg.  |
> |-------------------|------:|-----:|------:|------:|------:|------:|------:|------:|------:|-----:|------:|------:|------:|------:|------:|-------|
> |    SPARF + NeuS   |  3.37 | 0.03 |  0.03 |  0.03 |  0.56 |  0.18 |  0.03 | 12.92 |  3.51 | 0.03 |  7.75 |  5.98 |  0.17 | 13.35 |  0.03 |  3.20 |
> | SPARF + Geo-GenuS |  1.59 | 0.03 |  0.03 |  0.03 |  0.03 |  0.03 |  0.03 |  0.03 |  4.18 | 0.03 |  0.03 |  0.29 |  0.03 |  0.03 |  0.03 |  0.43 |
> |      SC-NeuS      |  0.03 | 0.50 |  0.03 |  0.03 |  0.03 |  0.04 |  0.08 |  0.14 |  0.11 | 0.26 |  0.06 |  0.25 |  0.04 |  0.13 |  0.10 |  0.12 |
> |        Ours       |  0.03 | 0.03 |  0.03 |  0.03 |  0.03 |  0.03 |  0.03 |  0.03 |  0.03 | 0.03 |  0.03 |  0.03 |  0.03 |  0.03 |  0.03 |  0.03 |
>
> For the translation error:
> |                   |    24 |    37 |    40 |    55 |    63 |    65 |    69 |    83 |    97 |   105 |   106 |   110 |   114 |   118 |   122 | Avg.  |
> |-------------------|------:|------:|------:|------:|------:|------:|------:|------:|------:|------:|------:|------:|------:|------:|------:|-------|
> |    SPARF + NeuS   | 13.41 |  0.98 |  0.69 |  0.39 |  2.44 |  1.36 |  1.15 | 13.02 | 10.37 | 10.37 |  9.52 | 14.27 |  1.78 | 15.28 |  0.60 |  6.37 |
> | SPARF + Geo-GenuS |  4.33 |  1.44 |  0.46 |  0.46 |  1.73 |  0.72 |  0.62 |  2.16 | 10.57 |  0.37 |  1.46 |  1.37 |  0.34 |  0.38 |  0.55 |  1.80 |
> |      SC-NeuS      |  0.97 |  3.43 |  0.78 |  0.33 |  0.60 |  0.12 |  0.21 |  0.67 |  0.20 |  0.85 |  0.15 |  0.42 |  0.13 |  0.33 |  0.33 |  0.63 |
> |        Ours       |  0.62 |  1.65 |  0.29 |  0.24 |  1.00 |  0.48 |  0.34 |  0.81 |  0.18 |  0.28 |  0.27 |  0.28 |  0.20 |  0.36 |  0.33 |  0.49 |
>
> For the Chamfer distance:
> |           |   24 |   37 |   40 |   55 |    63 |   65 |   69 |   83 |   97 |  105 |  106 |  110 |  114 |  118 |  122 | Avg. |
> |:-----------------:|-----:|-----:|-----:|-----:|------:|-----:|-----:|-----:|-----:|-----:|-----:|-----:|-----:|-----:|-----:|------|
> |    SPARF + NeuS   | 6.12 | 4.55 | 3.02 | 0.89 |  4.78 | 5.26 | 1.81 | 8.02 | 2.92 | 2.92 | 7.01 | 6.46 | 0.93 | 6.14 | 2.57 | 4.23 |
> | SPARF + Geo-GenuS | 4.95 | 5.52 | 3.42 | 0.99 |  4.57 | 3.73 | 1.87 | 2.49 | 3.29 | 1.82 | 3.27 | 2.48 | 0.93 | 2.02 | 2.54 | 2.93 |
> |      SC-NeuS      | 3.68 | 4.78 | 4.15 | 0.96 |  3.73 | 5.01 | 1.72 | 2.64 | 1.91 | 2.00 | 1.70 | 1.67 | 0.67 | 1.91 | 2.19 | 2.58 |
> |        Ours       | 2.90 | 4.26 | 2.91 | 0.84 |  3.63 | 3.51 | 1.47 | 2.86 | 1.31 | 1.67 | 1.98 | 1.70 | 0.57 | 1.65 | 1.69 | 2.20 |
>
> The comparative results validate that Geo-NeuS (integrated as SPARF + Geo-NeuS) successfully enhances the performance of the baseline (SPARF + NeuS) by leveraging its multi-view geometric constraints. However, its effectiveness decreases under our harsh testing conditions. The results show that both SC-NeuS and our method achieve superior stability and accuracy across all three metrics compared to SPARF + Geo-NeuS. Our approach, which explicitly utilizes uncertainty-aware occlusion handling, achieves the best overall performance, confirming that our mechanism for handling ambiguity is more robust than the geometric constraints employed by SPARF + Geo-NeuS.
>
> [1] Fu, Qiancheng, et al. "Geo-neus: Geometry-consistent neural implicit surfaces learning for multi-view reconstruction." Advances in Neural Information Processing Systems 35 (2022): 3403-3416.

---

### Official Review · Reviewer_ivGr · 2025-11-02

**Soundness:** 2
**Presentation:** 2
**Contribution:** 2
**Rating:** 2
**Confidence:** 5

**Summary:**

The paper tackles sparse-view implicit surface reconstruction when camera poses are noisy and viewpoints are widely separated. It proposes a joint optimization framework that (1) injects geometry-based uncertainty into pose refinement and geometric learning; and (2) introduces a pSDF regularization term for smoothing surfaces without over-blurring details. Experiments on DTU and BlendedMVS demonstrate state-of-the-art results in both pose and surface accuracy and geometry.

**Strengths:**

1. Satisfactory experimental results. The method conduct experiments on DTU, BlendedMVS datasets in both narrow and wide settings, and show clear advantages in pose estimation and surface reconstruction.

2. The writing is clear and the methodology is easy to reproduce.

3. The use of multi-view depth discrepancies of matched pairs to quantify uncertainty is insightful.

**Weaknesses:**

1. Limited technical novelty. In the Occlusion Handling part, the main contribution is using the projected depth discrepancy of 2D match pairs as an uncertainty estimate. The NCC loss and geometric consistency constraints are adaptations of existing works (e.g., NeuralWarp, PGSR). Additionally, the surface smoothing strategy is not substantially different from the constraint in iSDF. It mainly swaps the supervision source from LiDAR points to estimated surface points.

2. Insufficient comparison with recent work. The experimental baselines stop at methods from 2024 or earlier, lacking evaluations against the latest approaches.

**Questions:**

In iSDF, the pseudo SDF at a query point is defined by its distance to sparse sensor points, which are typically reliable, making the supervision well-founded. In this paper’s setting, however, the reference surface is coarse and may deviate from the true geometry. Using this as the SDF target could introduce bias and drive the optimization toward incorrect minima. The authors should discuss this risk or provide empirical evidence and sensitivity analysis.

---

> ### Author Response · Authors · 2025-11-13
>
> Thank you for the precious comments by reviewer ivGr. We fully acknowledge that our contribution does not lie in introducing novel loss functions, but in the innovative integration and geometry-aware design that ensures robust joint optimization.
>
> Instead of relying on standard 2D confidence or uniformly weighting patches in prior NCC-based pose refinement, we utilize a geometry-centric, 3D uncertainty $u$ both consistency losses. This strategy shows that jointly applying photometric and geometric consistency with the proposed uncertainty improves stability for large baselines. The key distinction is that our $u$ is a render-aware metric derived from the current SDF-rendered depth, coupling the uncertainty directly with the evolving implicit geometry. This makes it dynamically effective at depth ambiguity, unlike the fixed uncertainty provided by external correspondence networks.
>
> While the underlying concept of our pSDF is inspired by iSDF, it constitutes a critical adaptation required for our specific sparse-view, noisy-pose setup. iSDF addresses supervision from reliable, external sensor data (e.g., LiDAR). In contrast, our pSDF stabilizes geometry using the not-perfect, self-estimated coarse geometry. Because our supervision source is imprecise, using it as a target introduces a significant risk of optimization bias and potential geometric collapse. Our core technical contribution is the robust scheduling mechanism tailored to mitigate this risk, a feature unnecessary in iSDF. This mechanism ensures pSDF acts as an essential initial geometric stabilizer, preventing geometry from collapsing due to pose errors without sacrificing final high-frequency detail.
>
> We agree that comparisons with recent works are essential. We will add key recent methods related to our approach, such as Sparis (published in 2025), to our analysis. (ETA: Nov. 25th)
>
> We acknowledge that the success of the entire pipeline is contingent upon the quality of the initial coarse geometry. A failed coarse estimation under severe noise or sparsity can indeed lead to geometric collapse. This is a key limitation; however, recent advanced matching networks can address this problem, as demonstrated in the appendix by presenting the comparison with MASt3R [2].
>
> [1] Wu, Yulun, et al. "Sparis: Neural implicit surface reconstruction of indoor scenes from sparse views." Proceedings of the AAAI Conference on Artificial Intelligence. Vol. 39. No. 8. 2025.
>
> [2] Leroy, Vincent, Yohann Cabon, and Jérôme Revaud. "Grounding image matching in 3d with mast3r." European Conference on Computer Vision. Cham: Springer Nature Switzerland, 2024.

---

> ### Author Response · Authors · 2025-11-24
>
> We performed additional experiments for a recent work, Sparis [1]. For fair comparison, we replaced its dense matcher RoMa [2] to the PDCNet, which was utilized in SPARF. The experiments were conducted on the DTU large-baseline setup, matching the scenario presented in Table 1 of the main manuscript. The results are as follows:
>
> For the rotation error:
> |  |    24 |   37 |    40 |    55 |    63 |    65 |    69 |    83 |    97 |  105 |   106 |   110 |   114 |   118 |   122 | Avg.  |
> |:--------------:|------:|-----:|------:|------:|------:|------:|------:|------:|------:|-----:|------:|------:|------:|------:|------:|-------|
> |   SPARF + NeuS   |  3.37 | 0.03 |  0.03 |  0.03 |  0.56 |  0.18 |  0.03 | 12.92 |  3.51 | 0.03 |  7.75 |  5.98 |  0.17 | 13.35 |  0.03 |  3.20 |
> |     SPARF + Sparis     |  4.62 | 8.87 |  0.03 |  2.99 |  7.04 | 18.44 |  6.56 | 17.41 |  1.44 | 0.03 |  0.03 |  3.05 |  0.03 |  0.03 |  7.43 |  5.20 |
> |     SC-NeuS    |  0.03 | 0.50 |  0.03 |  0.03 |  0.03 |  0.04 |  0.08 |  0.14 |  0.11 | 0.26 |  0.06 |  0.25 |  0.04 |  0.13 |  0.10 |  0.12 |
> |      Ours      |  0.03 | 0.03 |  0.03 |  0.03 |  0.03 |  0.03 |  0.03 |  0.03 |  0.03 | 0.03 |  0.03 |  0.03 |  0.03 |  0.03 |  0.03 |  0.03 |
>
> For the translation error:
> |               |    24 |    37 |    40 |    55 |    63 |    65 |    69 |    83 |    97 |   105 |   106 |   110 |   114 |   118 |   122 | Avg.  |
> |---------------|------:|------:|------:|------:|------:|------:|------:|------:|------:|------:|------:|------:|------:|------:|------:|-------|
> |   SPARF + NeuS   | 13.41 |  0.98 |  0.69 |  0.39 |  2.44 |  1.36 |  1.15 | 13.02 | 10.37 | 10.37 |  9.52 | 14.27 |  1.78 | 15.28 |  0.60 |  6.37 |
> |     SPARF + Sparis    |  9.67 | 22.61 |  0.50 |  8.49 | 30.57 | 27.55 | 11.66 | 72.41 |  4.85 |  0.38 |  0.48 |  8.82 |  0.14 |  0.21 | 11.35 | 13.98 |
> |    SC-NeuS    |  0.97 |  3.43 |  0.78 |  0.33 |  0.60 |  0.12 |  0.21 |  0.67 |  0.20 |  0.85 |  0.15 |  0.42 |  0.13 |  0.33 |  0.33 |  0.63 |
> |      Ours     |  0.62 |  1.65 |  0.29 |  0.24 |  1.00 |  0.48 |  0.34 |  0.81 |  0.18 |  0.28 |  0.27 |  0.28 |  0.20 |  0.36 |  0.33 |  0.49 |
>
> For the Chamfer Distance:
> |           |   24 |   37 |   40 |   55 |    63 |   65 |   69 |   83 |   97 |  105 |  106 |  110 |  114 |  118 |  122 | Avg. |
> |:-----------------:|-----:|-----:|-----:|-----:|------:|-----:|-----:|-----:|-----:|-----:|-----:|-----:|-----:|-----:|-----:|------|
> |    SPARF + NeuS   | 6.12 | 4.55 | 3.02 | 0.89 |  4.78 | 5.26 | 1.81 | 8.02 | 2.92 | 2.92 | 7.01 | 6.46 | 0.93 | 6.14 | 2.57 | 4.23 |
> |   SPARF + Sparis  | 5.20 | 6.16 | 6.07 | 6.25 | -     | 7.95 | -    | -    | 2.77 | 1.13 | 3.98 | 3.99 | 0.82 | 2.17 | 6.53 | 4.42 |
> |      SC-NeuS      | 3.68 | 4.78 | 4.15 | 0.96 |  3.73 | 5.01 | 1.72 | 2.64 | 1.91 | 2.00 | 1.70 | 1.67 | 0.67 | 1.91 | 2.19 | 2.58 |
> |        Ours       | 2.90 | 4.26 | 2.91 | 0.84 |  3.63 | 3.51 | 1.47 | 2.86 | 1.31 | 1.67 | 1.98 | 1.70 | 0.57 | 1.65 | 1.69 | 2.20 |
>
> The quantitative results demonstrate that even the recent surface reconstruction approach designed for sparse views struggles in our harsh scenario. While methods like Sparis utilize strong dense geometric priors (surface normal), their reliance on a strong initial match makes them vulnerable to failure, as evidenced by the high average errors and cases of geometric collapse ('-'). This failure is attributed to the lack of a robust pose refinement process to correct initial errors during optimization. In contrast, our method, which incorporates a uncertainty-aware refinement mechanism, achieves near-ground-truth rotation and superior translation accuracy across all scenarios. This comparison highlights the necessity of an our uncertainty-aware refinement mechanism and stable learning for surfaces to stabilize both pose and geometry against failure in severely challenged environments.
>
> [1] Wu, Yulun, et al. "Sparis: Neural implicit surface reconstruction of indoor scenes from sparse views." Proceedings of the AAAI Conference on Artificial Intelligence. Vol. 39. No. 8. 2025.
>
> [2] Edstedt, Johan, et al. "Roma: Robust dense feature matching." Proceedings of the IEEE/CVF Conference on Computer Vision and Pattern Recognition. 2024.

---

> > ### Comment · Reviewer_ivGr · 2025-11-27
> >
> > Thank you for the detailed and insightful rebuttal. However, I still think the usage of occlusion handling and the proposed pSDF regularization to be incremental and lacking in novelty.
> >
> > In addition, I noticed that many of the compared baselines, such as NeuS and MonoSDF, are designed for dense-view settings. These methods are known to degrade under sparse-view conditions, which undermines the fairness of the comparison.
> >
> > Moreover, Sparis is designed for sparse indoor scene scenarios, and its effectiveness has not been validated on DTU, thus making its behavior in this benchmark less conclusive. Why not consider comparing with more recent sparse-view reconstruction methods which was directly conducted on DTU, such as FatesGS[1] or MAtCha[2]?
> >
> > [1]. Huang H, Wu Y, Deng C, et al. FatesGS: Fast and accurate sparse-view surface reconstruction using gaussian splatting with depth-feature consistency. Proceedings of the AAAI Conference on Artificial Intelligence. 2025, 39(4): 3644-3652.
> >
> > [2]. Guédon A, Ichikawa T, Yamashita K, et al. MAtCha Gaussians: Atlas of Charts for High-Quality Geometry and Photorealism From Sparse Views. Proceedings of the Computer Vision and Pattern Recognition Conference. 2025: 6001-6011.

---

> > > ### Author Response · Authors · 2025-11-28
> > >
> > > Thank you for suggesting FatesGS and MAtCha-Gaussians. The core distinction lies in the problem domain of noisy pose refinement. Both suggested works, which are based on 2D Gaussian Splatting (2DGS), are fundamentally limited in handling substantial initial pose noise. The extremely fast, localized nature of the 2DGS optimization process makes it fall into local optima when initial poses are far from the ground truth.
> > >
> > > Our empirical observations confirm this limitation:
> > >
> > >  Failure Mode (Artifacts): Our internal experiments attempting co-optimization under noisy poses resulted in local optimization failures, creating extraneous surface layers corresponding to the number of individual cameras. This phenomenon validates that standard 2D Gaussian Splatting (2DGS) is not robust to significant pose inaccuracies. Consequently, most successful 2DGS works must rely on strong initialization (i.e., highly accurate initial poses) rather than relying on the 2DGS optimization process itself to perform large pose corrections.
> > >
> > >  Required Initial Pose Quality: We observed that MASt3R initialization, which was used in MAtCha-Gaussians on the DTU dataset generally produces poses with minimal error.
> > >
> > > With these observations, we think that robust pose co-optimization in 2DGS requires either: (1) near-perfect initial poses (which is not our problem setting) or (2) specialized mechanisms dedicated to robustly handling pose noise.
> > >
> > > We initially excluded FatesGS and MAtCha because they lack this dedicated mechanism (2). We believe their excellent results from strong initialization (1). However, if the reviewer request these comparisons, we will run these experiments during the remaining rebuttal period and will report the results in a follow-up comment.

---

### Official Review · Reviewer_GieM · 2025-11-02

**Soundness:** 3
**Presentation:** 3
**Contribution:** 3
**Rating:** 6
**Confidence:** 4

**Summary:**

Summary: This paper proposes a method for robust surface reconstruction from spare and noisy camera poses with wide camera baselines. The core of the approach is two regularization strategies which help in generating faithful surfaces in sparse scenarios.

The first one is enforcing uncertainty aware geometric and photometric consistency as a loss to handle occlusions, where the uncertainty is pre-computed using pixel correspondences obtained from a pre-trained model. With this precomputed uncertainty the original photometric consistency loss proposed in [1] and geometric loss proposed in [2] are scaled appropriately to handle occlusions.

Secondly, a surface smoothness constraint is enforced, so that the representation is able to reach a solution even in sparse and degenerate scenarios.

Comparisons of both surface reconstruction quality and camera pose error on DTU and BlendedMVS dataset show that this method can robustly reconstruct the surface along with optimizing camera poses under sparse images and noisy pose as input.

**Strengths:**

* The motivation of the paper is clear, and the paper is well-written with clear presentation and the proposed method is easy to follow with adequate visual illustrations at the required places (Fig. 3, 4, 5).

* The paper is solving a relevant problem. Reconstructing a high fidelity surface from sparse and noisy input in a per-scene optimization framework is still a relevant problem as most of the recent methods focus on training a prior model on a dataset, because of which they are generally limited by the generalization ability compared to per-scene optimization techniques.

* The SDF smoothness constraint and the occlusion handling approach seems interesting and its effectiveness is validated with adequate ablations. Particularly the training strategies used to overcome local minima (over-smooth surfaces) is interesting.

**Weaknesses:**

* Although the strategy to robustly optimize for surfaces under the proposed constraints is interesting, the constraints itself are not new. Basically NCC loss and the geometric constraint is pretty common and all this technique does is scaling the loss with the precomputed uncertainty obtained from the correspondence network. Also the SDF regularization technique seems to be already proposed in iSDF [3] in which the authors propose a variant in which the optimization starts from a coarse geometry obtained after certain iterations.

* The paper uses a correspondence prediction network PDCNet [4] which seems to be pretty old as many advanced models like Dust3r [5] exist, which the authors themselves have mentioned in the Conclusion section (Sec. 5). Have they tried their approach with more latest models? I am asking this question, because I have a feeling that the ability of this method to handle wide baseline cameras is generally limited by the correspondence network’s ability for the same. An experiment to what extent this method can handle “wide” baselines is required to understand the robustness of this approach.

* Although the proposed approach for SDF regularization seems effective, have the authors tried by incorporating simple smoothness priors like depth from a foundation model into their framework? An ablation with such prior will be helpful to better understand the effectiveness of the smoothness constraint.

**Questions:**

Please refer to weakness.

Minor Questions:

B in eq. 6 is the batch size of what? Rays?

The paper should discuss some prior based methods which handle such wide baselines camera for reconstruction in the related work for e.g. papers like [6] and [7] and many more exists which solve similar problem with a prior based approach.

References:

[1] Shi-Sheng Huang, Zixin Zou, Yichi Zhang, Yan-Pei Cao, and Ying Shan. Sc-neus: Consistent neural surface reconstruction from sparse and noisy views. In Proceedings of the AAAI conference on artificial intelligence, volume 38, pp. 2357–2365, 2024c.

[2] Prune Truong, Marie-Julie Rakotosaona, Fabian Manhardt, and Federico Tombari. Sparf: Neural radiance fields from sparse and noisy poses. In Proceedings of the IEEE/CVF Conference on Computer Vision and Pattern Recognition, pp. 4190–4200, 2023.

[3] Joseph Ortiz, Alexander Clegg, Jing Dong, Edgar Sucar, David Novotny, Michael Zollhoefer, and Mustafa Mukadam. isdf: Real-time neural signed distance fields for robot perception. arXiv preprint arXiv:2204.02296, 2022.

[4] Prune Truong, Martin Danelljan, Luc Van Gool, and Radu Timofte. Learning accurate dense correspondence and when to trust them. In Proceedings of the IEEE/CVF conference on computer vision and pattern recognition, pp. 5714–5724, 2021.

[5] Shuzhe Wang, Vincent Leroy, Yohann Cabon, Boris Chidlovskii, and Jerome Revaud. Dust3r: Geometric 3d vision made easy. In Proceedings of the IEEE/CVF Conference on Computer Vision and Pattern Recognition, pp. 20697–20709, 2024.

[6] Vora, Aditya, Akshay Gadi Patil, and Hao Zhang. "Divinet: 3d reconstruction from disparate views using neural template regularization." Advances in Neural Information Processing Systems 36 (2023): 66768-66781.

[7] Du, Yilun, et al. "Learning to render novel views from wide-baseline stereo pairs." Proceedings of the IEEE/CVF Conference on Computer Vision and Pattern Recognition. 2023.

---

> ### Author Response · Authors · 2025-11-13
>
> Thank you for the precious comments by reviewer GieM. We fully acknowledge that our contribution is not in introducing novel loss functions, but in the novel integration and geometry-aware design that enables robust joint optimization.
>
> W1. Our novelty lies in how we compute and utilize the uncertainty to stabilize pose refinement in joint optimization.
>
> Unlike prior works that use generic 2D confidence or entropy, our method computes a geometry-centric uncertainty $u$ as the min-max normalized 3D discrepancy of back-projected correspondences, using the current SDF-rendered depth. This mechanism directly couples the uncertainty with the evolving implicit surface, making it highly effective at detecting occlusion and depth ambiguity specific to wide baselines.
>
> The uncertainty from correspondence networks (e.g., PDCNet) is fixed and only provided for the predicted matching points. In contrast, our render-aware 3D discrepancy is available to compute for arbitrary samples and dynamically changes during optimization, which is essential when jointly optimizing camera poses. Running a separate network to obtain confidence for arbitrary points would introduce significant computational overhead. (The detailed supporting evidence for this statement will be presented in a follow-up comment.)
>
> Prior NCC-based pose refinement typically uses only 2D confidences or treats all patches equally. We show that joint optimization with photometric and geometric consistency with the 3D uncertainty improves stability for large baselines.
>
> While the fundamental concept is inspired by distance-to-surface points for SDF supervision, our progressive SDF (pSDF) is a critical variant required for our challenging setup. iSDF addresses the challenge of reconstructing surfaces from reliable, external sensor points (e.g., LiDAR). In contrast, our pSDF is designed for a scenario where camera poses are noisy, views are sparse, and the supervision source is the unreliable, self-estimated coarse geometry. Using this unreliable geometry as a target introduces a significant risk of optimization bias and potential geometric collapse.
>
> Our core technical contribution is the robust scheduling mechanism to mitigate this risk, which is unnecessary for iSDF. This mechanism ensures that the loss acts as an initial geometric stabilizer essential for success in sparse-view and noisy pose refinement, preventing the geometry from collapsing due to pose errors without sacrificing the final high-frequency details.
>
> W2. We implemented an additional experiment using MASt3R [1] (an advanced version of Dust3r) for pose initialization, followed by optimization using InstantSplat [2] with 2DGS [3] with learnable camera parameters. As shown in Table 7 (Appendix), the advanced models did not precisely correct the initial camera poses. However, by adapting our contribution, we observe that the Chamfer distance is enhanced (2.33→2.24) and also for camera rotation error (0.52→0.45). This demonstrates that our method is effective as a robust post-optimization step, regardless of the initial pose estimation source.
>
> W3. We agree that incorporating strong monocular priors is a promising direction for future work to advance performance further. Directly applying metric-depth estimation results introduces a risk of double-layer surfaces or geometric conflicts if the external depth prior is not perfectly accurate. Therefore, methods that incorporate geometric cues softly, such as MonoSDF [4], would be more appropriate integration points.
>
> We are currently expanding our experimental validation by testing integration with existing soft-cue approaches: SPARF + MonoSDF [4] and SPARF + Sparis [5]. We hypothesize that these soft priors, which rely on external network prediction, may struggle to provide sufficient constraint in large-baseline scenarios. Such environments often demand harder, more precise geometric cues which are exceptionally difficult to obtain accurately. We will update the results as a comment once these comparison experiments are finalized (ETA - November 25st).
>
> [1] Leroy, Vincent, Yohann Cabon, and Jérôme Revaud. "Grounding image matching in 3d with mast3r." European Conference on Computer Vision. Cham: Springer Nature Switzerland, 2024.
>
> [2] Fan, Zhiwen, et al. "Instantsplat: Unbounded sparse-view pose-free gaussian splatting in 40 seconds." arXiv preprint arXiv:2403.20309 2.3 (2024): 4.
>
> [3] Huang, Binbin, et al. "2d gaussian splatting for geometrically accurate radiance fields." ACM SIGGRAPH 2024 conference papers. 2024.
>
> [4] Yu, Zehao, et al. "Monosdf: Exploring monocular geometric cues for neural implicit surface reconstruction." Advances in neural information processing systems 35 (2022): 25018-25032.
>
> [5] Wu, Yulun, et al. "Sparis: Neural implicit surface reconstruction of indoor scenes from sparse views." Proceedings of the AAAI Conference on Artificial Intelligence. Vol. 39. No. 8. 2025.

---

> ### Author Response · Authors · 2025-11-13
>
> Minor weakness: We apologize for the confusion regarding the batch notation in Section 3.3 and thank the reviewer for their precise and critical observation. This is a crucial distinction, and we will correct the notation immediately.
>
> We confirm that B denotes the batch size of sampled rays (scalar), while $\mathcal{B}$ denotes the set of correspondence pairs obtained from PDCNet.
>
> To accurately reflect our proposed method's independence from the correspondence network, we will remove the set $\mathcal{B}$ from the core uncertainty calculation (Eq. 5, Line 350).
>
> We will redefine the point $\text{q}$ as the geometrically projected point, where $\text{q} \leftarrow \pi(\pi^{-1}(\text{p}, z_\text{p}, P_t), P_s)$.
> Furthermore, we will clarify the matching correspondence loss in Eq. 10 by replacing $(\text{p},\text{q}) \in \mathcal{B}$ with $(\text{p}', \text{q}') \in \mathcal{B}$ to explicitly denote that these are the predicted correspondences from PDCNet.
> These changes establish that our uncertainty metric $u$ is based on geometric consistency (derived from rendered depths and projected points), not the confidence map of a pre-matched correspondence from PDCNet.
>
> So, we will modify the definition of the set $\mathcal{D}$ as: {$\lVert\pi^{-1}(\text{p},z_{\text{p}},P_t) - \pi^{-1}(\text{q},z_{\text{q}},P_s)
> \rVert_2^2$ $|$ $\text{q}\leftarrow\pi(\pi^{-1}(\text{p},z_\text{p},P_t),P_s))$}.
>
> Regarding the works mentioned (e.g., [6]), we offer the following clarification: While we acknowledge the existence of the suggested work [6], a direct comparison is hindered by the lack of public code. However, we demonstrate the strength of our approach by showing superior performance against established, open-sourced baselines like MonoSDF in surface reconstruction quality (as shown in Table 2). Furthermore, a review of the reported results in the main table of the DiviNet paper [6] indicates that our current method achieves exceeds their publicly reported metrics on key DTU scans. We believe this validates our approach against the state-of-the-art landscape.
>
> We are preparing to report comparisons with more recent works, as discussed in our previous comment.
>
> We note that suggested comparison work, [7], primarily focus on Novel View Synthesis (NVS) rather than high-fidelity implicit surface reconstruction, which is the main objective of our paper. Adapting NVS-centric methods to our surface reconstruction pipeline often requires significant, non-trivial engineering efforts and potentially changes to the core loss structure (e.g., integrating SDF constraints). We request the reviewers to consider the complex environment and the clear distinction between our geometry-focused objective and the NVS goal of the referenced work.
>
> [6]  Vora, Aditya, Akshay Gadi Patil, and Hao Zhang. "Divinet: 3d reconstruction from disparate views using neural template regularization." Advances in Neural Information Processing Systems 36 (2023): 66768-66781.
>
> [7] Du, Yilun, et al. "Learning to render novel views from wide-baseline stereo pairs." Proceedings of the IEEE/CVF Conference on Computer Vision and Pattern Recognition. 2023.

---

> ### Author Response · Authors · 2025-11-24
>
> We performed additional experiments for two works which are utilized soft geometric priors: SPARF + MonoSDF [4] (utilizing predicted surface normal and monocular depth) and SPARF + Sparis [5] (utilizing predicted surface normal). These experiments were conducted on the DTU large-baseline setup, matching the scenario presented in Table 1 of the main manuscript. The results are as follows:
>
> For the rotation error:
> |  |    24 |   37 |    40 |    55 |    63 |    65 |    69 |    83 |    97 |  105 |   106 |   110 |   114 |   118 |   122 | Avg.  |
> |:--------------:|------:|-----:|------:|------:|------:|------:|------:|------:|------:|-----:|------:|------:|------:|------:|------:|-------|
> |   SPARF + NeuS   |  3.37 | 0.03 |  0.03 |  0.03 |  0.56 |  0.18 |  0.03 | 12.92 |  3.51 | 0.03 |  7.75 |  5.98 |  0.17 | 13.35 |  0.03 |  3.20 |
> |     SPARF + MonoSDF    |  1.74 | 0.38 |  0.03 |  0.03 |  1.38 |  0.03 |  0.03 |  0.53 |  3.44 | 0.03 |  0.03 |  3.96 |  0.03 |  7.26 |  0.03 |  1.26 |
> |     SPARF + Sparis     |  4.62 | 8.87 |  0.03 |  2.99 |  7.04 | 18.44 |  6.56 | 17.41 |  1.44 | 0.03 |  0.03 |  3.05 |  0.03 |  0.03 |  7.43 |  5.20 |
> |     SC-NeuS    |  0.03 | 0.50 |  0.03 |  0.03 |  0.03 |  0.04 |  0.08 |  0.14 |  0.11 | 0.26 |  0.06 |  0.25 |  0.04 |  0.13 |  0.10 |  0.12 |
> |      Ours      |  0.03 | 0.03 |  0.03 |  0.03 |  0.03 |  0.03 |  0.03 |  0.03 |  0.03 | 0.03 |  0.03 |  0.03 |  0.03 |  0.03 |  0.03 |  0.03 |
>
> For the translation error:
> |               |    24 |    37 |    40 |    55 |    63 |    65 |    69 |    83 |    97 |   105 |   106 |   110 |   114 |   118 |   122 | Avg.  |
> |---------------|------:|------:|------:|------:|------:|------:|------:|------:|------:|------:|------:|------:|------:|------:|------:|-------|
> |   SPARF + NeuS   | 13.41 |  0.98 |  0.69 |  0.39 |  2.44 |  1.36 |  1.15 | 13.02 | 10.37 | 10.37 |  9.52 | 14.27 |  1.78 | 15.28 |  0.60 |  6.37 |
> |     SPARF + MonoSDF   |  5.51 |  4.88 |  0.54 |  0.48 |  3.07 |  0.68 |  0.37 |  3.39 | 10.43 |  0.70 |  0.98 |  9.30 |  0.52 |  7.35 |  0.26 |  3.23 |
> |     SPARF + Sparis    |  9.67 | 22.61 |  0.50 |  8.49 | 30.57 | 27.55 | 11.66 | 72.41 |  4.85 |  0.38 |  0.48 |  8.82 |  0.14 |  0.21 | 11.35 | 13.98 |
> |    SC-NeuS    |  0.97 |  3.43 |  0.78 |  0.33 |  0.60 |  0.12 |  0.21 |  0.67 |  0.20 |  0.85 |  0.15 |  0.42 |  0.13 |  0.33 |  0.33 |  0.63 |
> |      Ours     |  0.62 |  1.65 |  0.29 |  0.24 |  1.00 |  0.48 |  0.34 |  0.81 |  0.18 |  0.28 |  0.27 |  0.28 |  0.20 |  0.36 |  0.33 |  0.49 |
>
> For the Chamfer Distance:
> |           |   24 |   37 |   40 |   55 |    63 |   65 |   69 |   83 |   97 |  105 |  106 |  110 |  114 |  118 |  122 | Avg. |
> |:-----------------:|-----:|-----:|-----:|-----:|------:|-----:|-----:|-----:|-----:|-----:|-----:|-----:|-----:|-----:|-----:|------|
> |    SPARF + NeuS   | 6.12 | 4.55 | 3.02 | 0.89 |  4.78 | 5.26 | 1.81 | 8.02 | 2.92 | 2.92 | 7.01 | 6.46 | 0.93 | 6.14 | 2.57 | 4.23 |
> |  SPARF + MonoSDF  | 5.39 | 5.72 | 3.16 | 1.82 |  5.32 | 5.11 | 2.12 | 2.65 | 3.17 | 1.70 | 4.74 | 5.75 | 0.94 | 6.56 | 2.05 | 3.75 |
> |   SPARF + Sparis  | 5.20 | 6.16 | 6.07 | 6.25 | -     | 7.95 | -    | -    | 2.77 | 1.13 | 3.98 | 3.99 | 0.82 | 2.17 | 6.53 | 4.42 |
> |      SC-NeuS      | 3.68 | 4.78 | 4.15 | 0.96 |  3.73 | 5.01 | 1.72 | 2.64 | 1.91 | 2.00 | 1.70 | 1.67 | 0.67 | 1.91 | 2.19 | 2.58 |
> |        Ours       | 2.90 | 4.26 | 2.91 | 0.84 |  3.63 | 3.51 | 1.47 | 2.86 | 1.31 | 1.67 | 1.98 | 1.70 | 0.57 | 1.65 | 1.69 | 2.20 |
>
> Note that for fair comparison, we replaced the dense matcher used in Sparis (RoMa [6]) with the PDCNet we utilized. The results indicate that the naive adaptation of geometric priors (monocular depth/surface normal) struggles to be effective in our harsh setup. This suggests that methods relying on external cues (like those from MonoSDF or Sparis) fail when the underlying pose estimation is unstable. Specifically, Sparis struggles to build proper geometry due to camera pose correction failures. Since Sparis lacks an refinement process to enhance poses when its matching network fails, the geometry collapses. This analysis suggests that for stably utilizing dense geometric priors in the challenging scenarios, a much softer adaptation mechanism is required. While our method provides a robust solution for pose refinement, the effective integration of external priors necessitates further development of techniques that can softly incorporate geometry without collapsing under pose noise.
>
> [4] Yu, Zehao, et al. "Monosdf: Exploring monocular geometric cues for neural implicit surface reconstruction." Advances in neural information processing systems 35 (2022): 25018-25032.
>
> [5] Wu, Yulun, et al. "Sparis: Neural implicit surface reconstruction of indoor scenes from sparse views." Proceedings of the AAAI Conference on Artificial Intelligence. Vol. 39. No. 8. 2025.
>
> [6] Edstedt, Johan, et al. "Roma: Robust dense feature matching." Proceedings of the IEEE/CVF Conference on Computer Vision and Pattern Recognition. 2024.

---

### Meta-Review · Area_Chair_LGr4 · 2026-01-07

**Summary:**

The paper proposes a framework for implicit surface reconstruction from sparse and noisy camera views, which is an important problem. They introduce an uncertainty-aware consistency loss and a regularization term to smooth the surface during optimization using the model's own coarse geometry estimates. Reviewers agree that the setting is practical and challenging, and the paper is well-written and the proposed uncertainty modeling is an insightful heuristic.

However, the consensu among reviewers is that the contributions are incremental. The uncertainty mechamism essentially re-weights standard geometric losses, and the regularization term is a minor variant of what was proposed in iSDF. Moreover, the strategy of using estimated coarse geometry to supervise the SDF is viewed as risky compared to methods using reliable external sensor data. Reviewers remained concerned that this approach introduces bias and risks optimization collapse, a concern the rebuttal did not fully alleviate. Finally, a major issue throughout the review process among multiple reviewers was the choice of baselines. Initial comparisons relied heavily on methods designed for dense views or unfair configurations. While the authors added comparisons to methods like Geo-NeuS and Sparis, reviewers (particularly ivGr) noted that the evaluation still missed critical recent methods designed for sparse views. Also, although the authors added the comparison to SPARF + Geo-NeuS, I don't think the experiment can adequately address the reviewer SsPa's concern, since the performance boost for that case is not as prominent as what they showed in the ablation in the paper for the proposed method/ The author did not analyze further along the line.

Based on everything above, I think the paper does not currently meet the bar for acceptance.

**Reviewer Concerns:**

The rebuttal has addressed most of the concerns of GieM, but I don't think they address all the concerns from ivGr and ssPa.

**Reviewer Scores:**

I think all three reviewers will probably keep the original rating (6, 2, 4).

---

### Decision · Program_Chairs · 2026-01-26

Reject